# Molecular basis of ParA ATPase activation by the CTPase ParB during bacterial chromosome segregation

Lucas Schnabel[1,10], Manuel Osorio-Valeriano[1,2,8,10] ✉, Cecilia Perez-Borrajero [3], Wieland Steinchen[4,5], Christopher-Nils Mais[4,5], Bernd Simon [3,9], Juri Hanßmann [1,2], Maria Thamm[1], Janosch Hennig [3,6], Gert Bange[4,5,7] & Martin Thanbichler [1,2,5] ✉

DNA segregation by bacterial ParABS systems is mediated by transient tethering interactions between nucleoid-bound dimers of the ATPase ParA and centromere (parS)-associated complexes of the clamp-forming CTPase ParB. The lifetime of these interactions is limited by the ParB-dependent activation of ParA ATPase activity. Here, we elucidate the functional interplay between ParA and ParB in the model bacterium *Myxococcus xanthus*. We demonstrate that the N-terminal ParA-binding motif of ParB associates with a conserved bipartite binding pocket at the ParA dimer interface, in a manner dependent on ParB clamp closure. Moreover, we show that ParB and non-specific DNA interact cooperatively with ParA and synergistically induce structural changes in its Walker A and Walker B motifs that correlate with the activation of ParA ATPase activity. These results advance our understanding of the mechanism underlying DNA transport by the ParABS system and may help to unravel the mode of action of related cargo-positioning systems.

The faithful segregation of genetic material during cell division is essential for all living organisms. In most bacteria, this process is initiated by ParABS DNA partitioning systems, which are encoded on chromosomes as well as many low-copy number plasmids[1–3]. The components of these systems are highly conserved and include the Walker-type ATPase ParA, the DNA-binding protein ParB, and a short palindromic DNA sequence called *parS*. One or several copies of *parS* are clustered close to the replication origin of target replicons, constituting the centromeric region that acts as the anchor point for the segregation machinery. ParB recognizes these sequences and, after initial specific binding, spreads into the neighboring DNA regions,

forming a large nucleoprotein complex that typically covers 10–20 kb of the origin region[4,5]. This structure, known as the partition complex, then interacts dynamically with the nucleoid-associated ATPase ParA, which promotes the segregation and proper positioning of sister replicons within the cell[4].

ParB is a homodimeric CTPase that acts as a DNA-sliding clamp. Its function is closely linked to its conserved domain architecture, which includes (1) an N-terminal ParB/Srx-like CTPase domain, (2) a central DNA-binding domain responsible for *parS* recognition, (3) an unstructured linker region, and (4) a C-terminal self-association domain that stably connects the two subunits of a ParB dimer[6,7]. When

[1]Department of Biology, Marburg University, Marburg, Germany. [2]Max Planck Fellow Group Bacterial Cell Biology, Max Planck Institute for Terrestrial Microbiology, Marburg, Germany. [3]Molecular Systems Biology Unit, European Molecular Biology Laboratory (EMBL), Heidelberg, Germany. [4]Department of Chemistry, Marburg University, Marburg, Germany. [5]Center for Synthetic Microbiology (SYNMIKRO), Marburg, Germany. [6]Chair of Biochemistry IV, Biophysical Chemistry, University of Bayreuth, Bayreuth, Germany. [7]Max Planck Fellow Group Molecular Physiology of Microbes, Max Planck Institute for Terrestrial Microbiology, Marburg, Germany. [8]Present address: Department of Cell Biology, Harvard Medical School, Boston, MA, USA. [9]Present address: Department of Molecular Biology and Biophysics, University of Connecticut Health Center, Farmington, CT, USA. [10]These authors contributed equally: Lucas Schnabel, Manuel Osorio-Valeriano. ✉e-mail: Manuel_OsorioValeriano@hms.harvard.edu; thanbichler@uni-marburg.de

the two DNA-binding domains of a ParB clamp interact with the inverted repeats of a *parS* site, the two ParB/Srx domains associate in a CTP-dependent manner, thereby closing the ParB clamp around the DNA molecule[8–10]. This process induces a conformational change in the DNA-binding domains that triggers their release from *parS* and displaces the DNA into a central opening in the closed ParB clamp. As a consequence, ParB starts to diffuse laterally on the DNA, thus making *parS* accessible again to other ParB dimers and enabling the next loading cycle[10–14]. After an extended period of random diffusion, CTP hydrolysis triggers the dissociation of the two ParB/Srx domains, which opens the DNA entry gate and allows the detachment of the ParB clamp from the DNA[11,12,15]. Repeated cycles of loading, sliding, and detachment thus lead to the establishment of a one-dimensional diffusion gradient of ParB clamps that originates at *parS* sites. After clamp opening, the ParB/Srx domains can also engage in bridging interactions that link distal ParB dimers and promote the *parS*-independent recruitment of additional clamps[13,14,16–19], which helps to further compact the partition complex and thus facilitate its translocation by the partitioning ATPase ParA.

ParA belongs to the ParA/MinD family of P-loop ATPases, a structurally conserved group of proteins that mediate the positioning of various macromolecular cargoes within bacterial and archaeal cells[20,21], including chromosomes and plasmids[2,4], the cell division apparatus[22–24], chemoreceptor arrays[25,26], carboxysomes[27,28] and the conjugation machinery[29]. As a common feature, the different family members share the ability to oscillate between a monomeric and dimeric state with distinct activities, driven by ATP binding and hydrolysis[20,30]. When associated with ATP, two monomers interact face-to-face, with the nucleotides sandwiched in between the two subunits. In the case of ParA, ATP-bound homodimers have non-specific DNA-binding activity[31–34], mediated by clusters of positively charged residues that line the edge of the subunit interface[31,35–37]. As a consequence, they associate randomly with the nucleoid[38–40], where they interact dynamically with ParB partition complexes to promote the segregation of sister replicons. DNA movement is thought to be driven by a ratchet-like mechanism in which partition complexes are passed on between adjacent mobile loops of chromosomal DNA, based on transient tethering interactions with nucleoid-associated ParA dimers[41–43]. The lifetime of these tethers is limited by the ATPase activity of ParA, which is triggered upon contact with its cognate ParB protein, leading to the dissociation of the dimeric complex[31,44–47]. The resulting ADP-bound monomers detach from both DNA and ParB and diffuse freely within the cytoplasm. Spontaneous nucleotide exchange then restores the ATP-bound state and thus enables the next cycle of dimerization and nucleoid binding. The time delay between nucleotide hydrolysis and re-dimerization prevents the formation of new ParA dimers in the immediate vicinity of the original binding site. This effect gives rise to a ParA-free depletion zone in the wake of segregating partition complexes that ensures their directional movement towards regions of high ParA dimer occupancy in the flanking nucleoid segments[32,41,43,48]. During plasmid partitioning, multiple segregation events can occur simultaneously, leading to the equidistant arrangement of plasmid copies along the length of the nucleoid[49,50]. Chromosome partitioning, by contrast, typically involves a single round of sister origin segregation along a polarized nucleoid-spanning ParA gradient[51–53], which may be reinforced through the sequestration of free ParA monomers by cell pole-associated scaffolding proteins[54–59].

The interaction of ParB with its cognate ParA protein is mediated by an N-terminal unstructured region preceding the ParB/Srx domain, which contains a conserved motif composed of aliphatic and positively charged residues[31,60–63]. Notably, a short synthetic peptide comprising this motif is sufficient to stimulate the ATPase activity of ParA in vitro, indicating that it contains all critical interaction determinants[31,64]. Previous crystallographic studies of the ParA homolog of plasmid TP228 suggested that the conserved N-terminal region of its cognate

centromere-binding protein (ParG) binds close to the edge of the dimer interface, with one of its arginine residues acting as an arginine finger that stimulates ATP hydrolysis by completing the active site of the ParA dimer[65,66]. However, this residue was located at a considerable distance from the bound nucleotide, calling into question its direct involvement in the catalytic mechanism. Other, spatially distinct interaction regions were suggested for the ATPase-stimulating peptides of ParA from *Pseudomonas aeruginosa*[67] and the ParA homolog δ of plasmid pSM19035[68], based on low-resolution chemical crosslinking and two-hybrid analyses. Finally, yet another ParB-binding site has recently been suggested by crystallographic studies of ParA from *Helicobacter pylori* associated with a short proteolytic fragment that contained part of the conserved N-terminal motif of ParB[69]. Due to the inherent difficulty of directly monitoring the interaction between ParA and ParB in vitro, it has proven difficult to experimentally verify the functional relevance of the observed interaction sites. The situation is further complicated by the fact that different aspects of the interaction between ParA and ParB have been studied in different model systems. Overall, therefore, despite the substantial amount of data collected in the field, a unified understanding of the mechanism that governs the interaction between these two proteins and mediates the ParB-dependent stimulation of the ParA ATPase activity remains elusive.

In this study, we comprehensively investigate the regulatory interplay between ParA and ParB in the model bacterium *Myxococcus xanthus*. We show that ParB clamps preferentially associate with ParA in the closed state, in a manner dependent on conserved hydrophobic residues in their N-terminal ParA-interaction motif, effectively restricting the ParA-ParB interaction to partition complexes. Moreover, we demonstrate that ParB binding stimulates both the DNA-binding and ATPase activities of ParA. A combination of computational modeling, nuclear magnetic resonance (NMR) spectroscopy and mutational studies reveals that ParB interacts with a conserved hydrophobic pocket at the edge of the ParA dimer interface that is formed by residues from both subunits and includes a loop immediately adjacent to the deviant Walker A motif of ParA. Using hydrogen-deuterium-exchange (HDX) mass spectrometry analysis, we then show that ParB and DNA binding act synergistically to induce conformational changes in the Walker A and Walker B motifs that correlate with their stimulatory effects on the ParA ATPase activity. Finally, we provide evidence that the central arginine/lysine residue in the ParA-binding motif of ParB interacts with a conserved region on the ParA surface adjacent to the primary ParB-binding pocket that is critically involved in stimulating the ParA ATPase activity. Collectively, these findings provide detailed insight into the molecular mechanism underlying the formation of the transient tethering interactions between ParB and ParA that drive the translocation of partition complexes during bacterial DNA segregation.

## Results

### ParA binding requires juxtaposition of the N-terminal peptide by closure of the ParB clamp

Previous work has shown that the conserved N-terminal peptide of ParB is necessary and sufficient for ParA binding[31,60,61]. However, it has been difficult to analyze the ParA-ParB interaction in more detail, because its assessment was typically based on ATPase assays as an indirect readout. To monitor the binding reactions more directly, we adapted a biolayer interferometry (BLI) assay that allows the reconstitution of model partition complexes in vitro[8]. To this end, both ends of a double-biotinylated double-stranded DNA fragment (234 bp) containing a central *parS* site were immobilized on a streptavidin-coated BLI biosensor and then loaded with *M. xanthus* ParB in the presence of the poorly hydrolyzable CTP analog CTPγS (Fig. 1a). The resulting stable nucleoprotein complexes[12] were then used as analytes to study the interaction of ParA with DNA-associated closed ParB clamps (Fig. 1b), under conditions that closely mimic the cellular

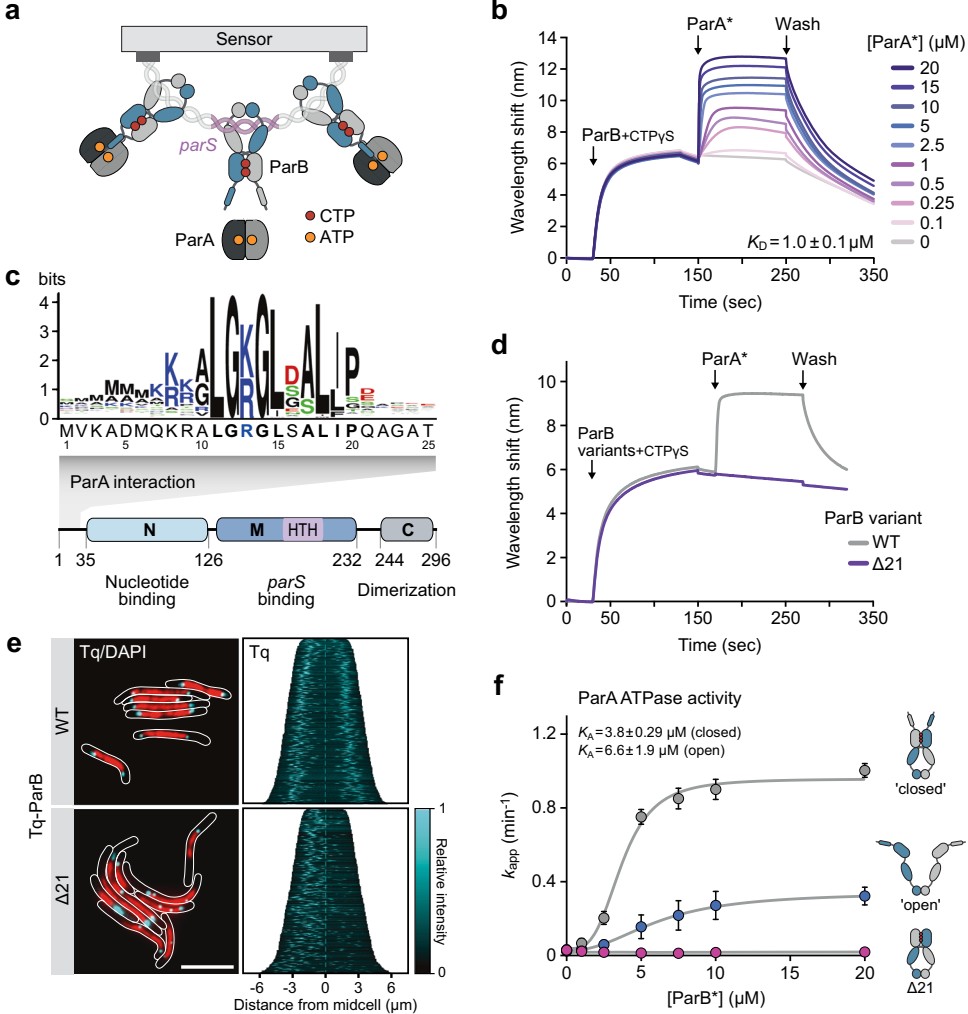

**Fig. 1 | ParA binds the N-terminal region of ParB with a preference for closed ParB clamps. a** Schematic of the biolayer interferometry (BLI) setup used to study the interaction of ParA with ParB. Double-biotinylated DNA fragments (234 bp) containing a central *M. xanthus parS* site were immobilized on a streptavidin-coated biosensor. Subsequently, ParB or its CTP hydrolysis-deficient ParB-Q52A variant (10 μM) was loaded onto the closed fragments in the presence of CTPγS or CTP (1 mM), respectively, to form stable model partition complexes, which were then analyzed for their interaction with ParA. **b** BLI analysis of the interaction of ParA with DNA-bound ParB. Wild-type ParB was loaded onto closed *parS*-containing DNA as described in panel a and probed with the indicated concentrations of ParA-R238E (ParA*) in the presence of ATP (1 mM). At the end of the association phase, the biosensor was transferred into protein- and nucleotide-free buffer (Wash) to follow the dissociation kinetics. Shown is a representative experiment (*n* = 3). **c** Conservation of the N-terminal region of ParB. Shown are a schematic depicting the domain organization of ParB and a sequence logo showing the conserved ParA-binding motif, based on an alignment of 3800 ParB homologs obtained by protein BLAST analysis with *M. xanthus* ParB as a query. Residues are colored according to their physico-chemical properties (black: hydrophobic, blue: positively charged, red: negatively charged, green: polar). **d** BLI analysis investigating the role of the N-terminal region of ParB in the ParA-ParB interaction. ParB or ParBΔ21 were loaded onto *closed parS*-containing DNA as described in (**a**) and probed with ParA-R238E (ParA*) (10 μM) in the presence of ATP and CTP (1 mM each). Shown is a representative experiment (*n* = 3 independent replicates). **e** Subcellular localization of

ParB and ParBΔ21 in *M. xanthus*. Cells producing Tq-ParB (MO072) or Tq-ParBΔ21 (LS007) in place of wild-type ParB were stained with 4′,6-diamidino-2-phenylindole (DAPI) prior to analysis by phase contrast and fluorescence microscopy. The images show overlays of the sfmTurquoise2ox (Tq) and DAPI signals, with the cell outlines indicated in white (bar: 5 μm). The demographs on the right summarize the sub-cellular distribution of the Tq signal in representative subpopulations of cells (*n* = 400 per strain). The single-cell fluorescence profiles were sorted according to cell length and stacked on top of each other. **f** Role of the opening state of ParB clamps in the stimulation of the ParA ATPase activity. The graphs show that ATPase activities of ParA (5 μM) incubated with ParB-Q52A (ParB*) (10 μM) in a buffer containing ATP (1 mM) and salmon sperm DNA (100 μg/mL) in the absence ("open") or presence ("closed") of CTP (1 mM) and a short *parS*-containing DNA stem-loop (250 nM). A reaction containing closed ParBΔ21 clamps served as a negative control (Δ21). Data represent the mean of three independent replicates (±SD). The results were fitted to a Hill equation to account for the sigmoidal shape of the binding curve. However, note that for the first data points, [ParB*] is within the range of the ParA concentration and close to the $K_D$ of the ParA-ParB interaction (see **b**), leading to titration of the free ligand species. Moreover, due to the close linkage of the N-terminal peptides in the ParB dimer, the reactions may be influenced by avidity effects. Therefore, it is not straightforward to determine whether the binding behavior observed actually signifies cooperativity in the binding process. Source data are provided as a Source data file.

environment. To avoid the association of ParA with the immobilized DNA and thus limit the measurements to ParB binding, the analysis was performed with a ParA variant (R238E, termed ParA*) that lacked non-specific DNA-binding activity[54] (Supplementary Fig. 1a) but still showed a normal interaction with ParB (Supplementary Fig. 1b). When

biosensors carrying preformed ParB-DNA complexes were probed with purified ParA*, we observed strong concentration-dependent binding with an apparent equilibrium dissociation constant ($K_D$) of ~1 μM (Fig. 1b and Supplementary Fig. 1c). ParA* binding was unde-tectable in the absence of ParB (Supplementary Fig. 1a) and markedly

reduced if ParB clamps were loaded with CTP and thus able to dissociate from the DNA during the measurement due to nucleotide hydrolysis (Supplementary Fig. 1d), confirming the specificity of the interaction for ParB. Notably, nucleotide content analysis revealed that ParA was, to a large extent, purified in the ATP-bound, dimeric state (Supplementary Fig. 2a,b), which suggests that it forms stable ATP-bound dimers without appreciable intrinsic ATPase activity (as confirmed below). Consistent with this notion, we found that purified ParA* associated with ParB-DNA complexes independently of the nature of the nucleotide added to the reaction buffer (Supplementary Fig. 2c). Together, these results demonstrate that ParA dynamically interacts with model partition complexes in vitro, in a process that does not require its non-specific DNA binding activity.

Next, we investigated the determinants of ParB mediating the interaction with ParA. Like its orthologs, *M. xanthus* ParB[12] features an unstructured N-terminal region (amino acids 1–35)[12] that contains a conserved ParA-binding motif (amino acids 11–19, Fig. 1c). To verify the role of this motif, we constructed an N-terminally truncated ParB variant lacking the first 21 amino acids (ParBΔ21) and assessed its ability to recruit ParA in vitro. As expected, the mutant protein was still able to form model partition complexes, but its ability to interact with ParA was completely abolished (Fig. 1d). To further validate this result, we determined the effect of the truncation on the localization pattern and functionality of ParB in vivo. To this end, wild-type ParB and ParBΔ21 were fused to the cyan fluorescent protein sfmTurquoise2ox (Tq) and produced in cells depleted of the endogenous ParB protein (Supplementary Fig. 3a). Microscopic analysis showed that the wild-type fusion protein was fully functional, as reflected by the fact that sister nucleoids and partition complexes were well-segregated and arranged symmetrically in the two halves of the cell (Fig. 1e and Supplementary Fig. 3b). Cells producing the truncated variant, by contrast, displayed severe chromosome segregation defects, with two or more partition complexes clustering in one cell half and many cell division events giving rise to anucleate offspring. Although this mutant phenotype was likely to be caused by the loss of tethering interactions between ParA and ParB, it was conceivable that the disruption of ParA binding also affected the integrity or dynamics of partition complexes. However, we did not observe any influence of ParA on the CTPase activity of ParB in vitro (Supplementary Fig. 4a). Moreover, ParA binding had no effect on the turnover dynamics of partition complexes in vivo (Supplementary Fig. 4b–d), suggesting that partition complex formation and segregation are functionally independent processes.

Previous work has shown that the interaction of ParB clamps with CTP and *parS* induces the homodimerization of their ParB/Srx domains, thereby closing the DNA entry gate and juxtaposing the unstructured N-terminal regions of the two subunits[9–12]. To test whether this structural rearrangement could facilitate the association of ParA with the conserved N-terminal interaction motif[19], we compared the ability of ParB clamps to stimulate the ParA ATPase activity in the open and closed state. This analysis was performed with a ParB variant lacking CTPase activity (ParB-Q52A)[12] to enable robust clamp closure in the presence of CTP and *parS* DNA. Consistent with previous work[19,44,46], elevated concentrations of open ParB-Q52A clamps led to a moderate increase in the rate of ATP hydrolysis when assayed in the presence of non-specific salmon sperm DNA (Fig. 1f). Notably, however, a considerably stronger effect was observed in reactions containing closed clamps, whereas stimulatory activity was undetectable for closed clamps of a ParB-Q52A variant lacking the unstructured N-terminal regions. The maximal turnover rates obtained were similar to those reported for other ParA orthologs[19,31,42]. These results demonstrate that clamp closure is critical for a productive interaction between ParB and ParA, likely by raising the local concentration of the N-terminal regions and thus ensuring that both ParA-binding motifs of a ParB clamp can simultaneously associate with a bound ParA dimer. As

a consequence, the ParA-ParB interaction is effectively limited to partition complexes in vivo.

## ParB binding stimulates the ATPase and DNA-binding activities of ParA

After verifying that ParB requires its N-terminal unstructured region to associate with ParA and stimulate its ATPase activity, we aimed to determine the mode of action of this key functional element. To this end, we first determined whether a short synthetic peptide containing the ParA-binding motif (ParB$_{1-20}$) was sufficient to achieve ParA binding. In line with previous studies[31,42,47], in vitro ATPase assays showed that the peptide hardly affected the ParA ATPase activity when added alone but showed a strong stimulatory effect if non-specific DNA was included in the reaction (Fig. 2a, b). These results demonstrate that the N-terminal peptide and non-specific DNA act cooperatively to induce a conformational change in ParA dimers that considerably improves their catalytic rate. Notably, although the maximal turnover rate achieved at saturating peptide concentrations was similar to that obtained with full-length ParB, the stimulatory effect of the peptide was more than tenfold lower than that of closed ParB clamps (compare Fig. 1h). This result further supports the notion that efficient stimulation of the ParA ATPase activity requires pairing of the N-terminal ParA-binding motifs. To further investigate the cooperativity of peptide and DNA binding to ParA, we performed BLI-based interaction assays analyzing the association of ParA with DNA-coated biosensors (Fig. 2c). Importantly, the presence of ParB$_{1-20}$ led to a strong (~20-fold) increase in the non-specific DNA-binding affinity of ParA (Fig. 2d and Supplementary Fig. 5a–c), as reported previously for the F plasmid ParA ortholog[19]. Similar results were obtained with a ParA variant (D60A) that lacks a catalytic aspartic acid residue coordinating the attacking nucleophilic water during the ATP hydrolysis reaction and is therefore devoid of ATPase activity[31–34] (as shown below). Thus, the observed increase in affinity was independent of the stimulatory effect of the peptide on ATP turnover (Fig. 2d and Supplementary Fig. 5d). Together, these findings demonstrate that ParB stimulates both the DNA-binding and ATPase activities of ParA. In this way, it may reinforce the ParA-mediated tethers between partition complexes and the nucleoid, while also priming these tethering complexes for disassembly.

## ParB interacts with a conserved hydrophobic pocket at the ParA dimer interface

To clarify the precise mode of interaction between ParB and ParA, we set out to determine the molecular structure of a ParA dimer in complex with ParB$_{1-20}$. We first attempted to determine the crystal structure of a ParA-D60A variant lacking the N- and C-terminal unstructured regions (ParA$_{21-274}$-D60A) in complex with ATP and the ParB$_{1-20}$ peptide. However, X-ray analysis revealed that all crystals lacked the peptide and only contained the ParA$_{21-274}$-D60A•ATP dimer. The structure of the dimeric complex, solved to a resolution of 1.6 Å (Supplementary Fig. 6a and Supplementary Table 1), was highly similar to that of the *H. pylori* ParA•ATP dimer[35] (RMSD of 0.767 Å for 225 paired C$_\alpha$ atoms), in line with the high functional conservation of the ParABS system among bacteria[4]. As an alternative approach to determine the ParB binding site, we computationally predicted the structure of a ParA$_{21-274}$ dimer in complex with a ParB dimer (Fig. 2e) or two ParB$_{1-20}$ peptides (Fig. 2f and Supplementary Fig. 6b) using AlphaFold-Multimer[70]. Consistent with our experimental data and previous findings[19], this analysis suggested that the interaction of ParB clamps with the ParA dimer involves the N-terminal tails of both ParB subunits. In the models obtained, these regions fold into short α-helices that interact with opposing edges of the symmetric ParA dimer interface, opposite the predicted DNA-binding site. Their association is predicted to rely on four conserved hydrophobic residues in the ParA-interaction motif (compare Fig. 1c) that are accommodated by a

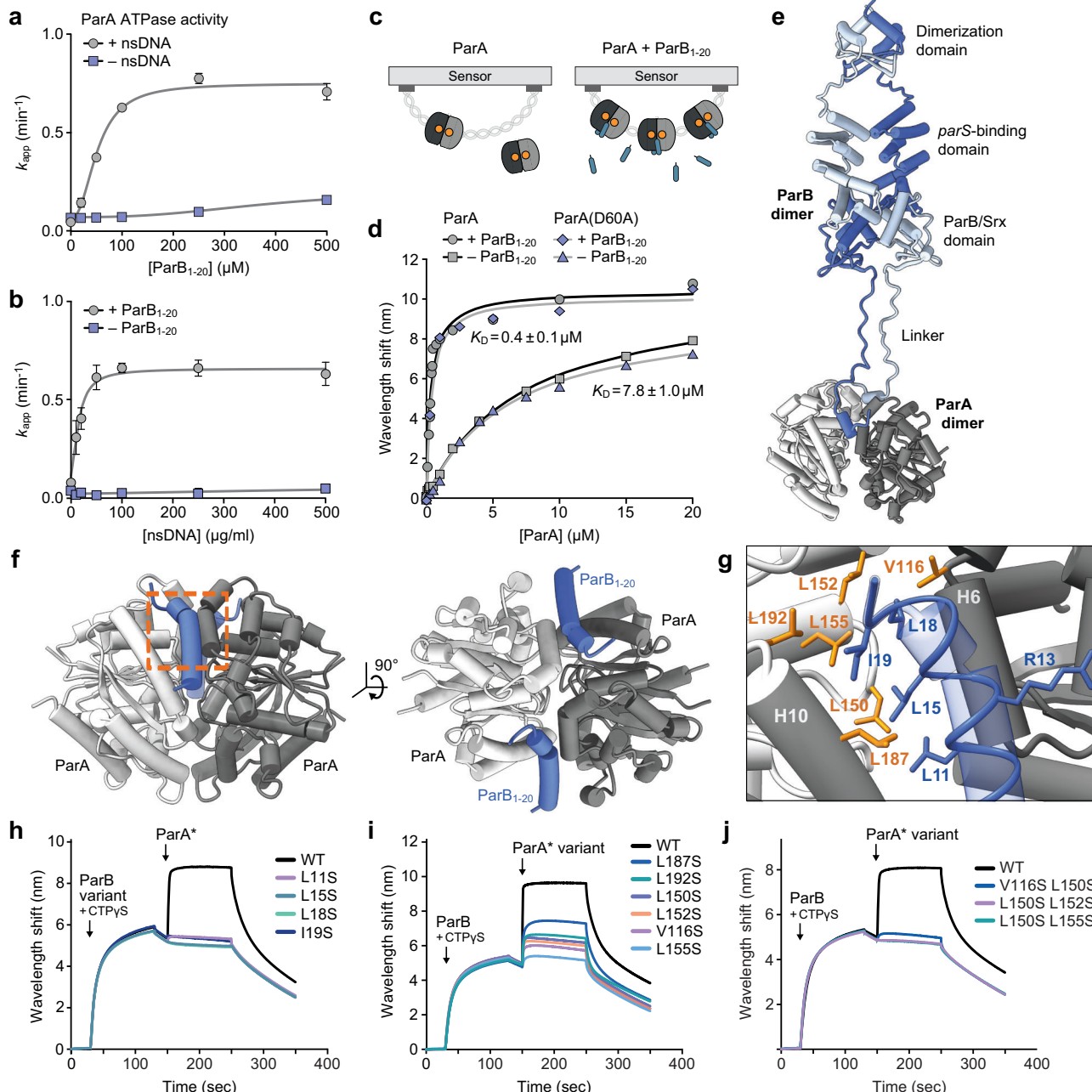

**Fig. 2 | The conserved N-terminal region of ParB interacts with a hydrophobic cleft at the ParA dimer interface. a** Stimulation of the ParA ATPase activity by the ParB$_{1-20}$ peptide. The graph shows the ATPase activity of ParA (5 μM) in reactions containing ATP (1 mM) and increasing concentrations of ParB$_{1-20}$ in absence and presence of salmon sperm DNA (100 μg/mL). Data represent the mean of three independent replicates (±SD). The results were fitted to a Hill equation. **b** Stimulation of the ATPase activity of ParA by DNA. The graph shows the ATPase activity of ParA (5 μM) in reactions containing ATP (1 mM) and increasing concentrations of salmon sperm DNA in the presence and absence of ParB$_{1-20}$ (100 μM). Data represent the mean of three independent replicates (±SD). The results were fitted to a Hill equation. **c** Schematic of the BLI setup used to study the non-specific interaction of ParA with DNA. **d** BLI analysis investigating the DNA-binding affinity of ParA in the presence of the ParB$_{1-20}$ peptide. Streptavidin-coated biosensors carrying a closed double-biotinylated DNA fragment (234 bp) were probed with the indicated concentrations of wild-type ParA in the absence or presence of ParB$_{1-20}$ (150 μM). The wavelength shift values obtained at the end of the association phase were plotted against the corresponding ParA concentrations and fitted to a one-site specific-binding model. The graph shows representative experiments. The $K_D$ values given in the graph indicate the mean (±SD) of three independent replicates. **e** Predicted structure of an *M. xanthus* ParA$_{21-274}$ (gray) dimer in complex with a

closed *M. xanthus* ParB clamp (blue), determined with Alphafold-Multimer[70] and shown in cartoon representation. **f** Predicted structure of an *M. xanthus* ParA$_{21-274}$ dimer (gray) in complex with two ParB$_{1-20}$ peptides (blue), determined with Alphafold-Multimer[70] and shown in cartoon representation. The orange rectangle indicates the region highlighted in (**f**). The structural coordinates are provided in Supplementary Data 2. **g** Predicted ParB-binding site, based on the structural model described in (**f**). Residues in ParA (orange) and ParB$_{1-20}$ (blue) that are involved in the interaction are shown in stick representation. **h** BLI analysis of the interaction of ParA with DNA-bound ParB variants carrying exchanges in the ParA-binding motif. The indicated ParB variants were loaded onto a closed *parS*-containing DNA fragment (see Fig. 1a) and probed with ParA-R238E (ParA*) (5 μM) in the presence of ATP and CTP (1 mM each). At the end of the association phase, the biosensors were transferred into nucleotide- and protein-free buffer to monitor the dissociation reactions. Shown are the results of a representative experiment (*n* = 2 independent replicates). **i, j** BLI analysis of the ParB-binding activity of ParA variants with one (**i**) or two (**j**) amino acid substitutions in the predicted ParB-binding pocket. Wild-type ParB was loaded onto a closed *parS*-containing DNA fragment (see Fig. 1a) and probed with the indicated ParA-R238E (ParA*) variants (5 μM) as described in panel h. Source data are provided as a Source data file.

hydrophobic pocket on the ParA dimer surface composed of residues from both ParA subunits (Fig. 2g). The structure of the ParA$_{21-274}$ dimer in these high-confidence models is similar to the crystal structure of the ParA$_{21-274}$-D60A•ATP dimer complex (RMSD of 0.527 Å for 242 paired C$_\alpha$ atoms) (Supplementary Fig. 6), supporting the general validity of the modeling approach.

To experimentally verify the binding interfaces suggested by the AlphaFold models, we generated variants of ParA and ParB that lacked putative interaction determinants. To this end, we replaced the hydrophobic residues L11, L15, L18, or I19 in ParB as well as V116, L150, L152, L155, L187, or L192 in ParA* (Fig. 2g) with the polar amino acid serine, thereby weakening the hydrophobic interactions that were predicted to mediate the association between ParA and ParB. Subsequently, we purified the mutant proteins and analyzed the effects of the different amino-acid exchanges on their interaction behavior, using the BLI setup described above (see Fig. 1a). Consistent with their high conservation score (Fig. 1e), all large hydrophobic residues in the ParA-binding motif of ParB emerged to be critical for the recruitment of ParA to model partition complexes in vitro, with each of the substitutions investigated strongly impairing the interaction (Fig. 2h). Similarly, the exchange of single hydrophobic residues in the predicted ParB-binding pocket of ParA was sufficient to markedly reduce the ParA-ParB interaction (Fig. 2i), and double substitutions in this region again largely abolished the association between the two proteins (Fig. 2j). Size-exclusion chromatography analysis of representative proteins confirmed that the amino acid exchanges did not have any noticable effects on ParA dimerization (Supplementary Fig. 7). Together, these findings strongly support the prediction that the ParA-binding motif of ParB associates with a hydrophobic pocket at the dimer interface of ParA, explaining why the interaction between the two proteins only occurs upon transition of ParA to the dimeric state.

To corroborate the results of the computational and mutational analyses, we aimed to map the ParB-binding pocket of ParA using NMR spectroscopy, a technique that is well-suited to study dynamic complexes in near-native conditions. For this purpose, we monitored the chemical shift perturbations (CSPs) in the amide signals of ATP-bound isotopically labeled ParA$_{21-274}$-D60A that were induced upon the addition of unlabeled ParB$_{1-20}$ peptide. Two-dimensional $^1$H-$^{15}$N heteronuclear single quantum correlation (HSQC) experiments yielded ~250 distinct chemical shifts (Supplementary Fig. 8), confirming that ParA dimers have a symmetrical architecture in solution, as previously observed in crystallographic studies[35,66]. Stepwise addition of ParB$_{1-20}$ resulted in a large number of CSPs in the intermediate exchange regime, characteristic of interactions with equilibrium dissociation constants in the 1–100 μM range[71], (Fig. 3a; see Supplementary Fig. 8 for the full spectra). Despite deuteration of the protein and the use of transverse relaxation-optimized spectroscopy (TROSY) experiments[72], we achieved only partial backbone assignments due to the large size of the ParA dimer. However, most residues exhibiting large CSPs corresponded to residues predicted to directly interact with ParB$_{1-20}$ (L150, G151, L152 and L192; compare Fig. 2g) or to lie in the immediate vicinity of the bound peptide (D110, T112, G113, E117, L118, G151, E176, L181, T184 and Q190) (Fig. 3b). In contrast, residues located at a distance from the predicted ParB-binding pocket exclusively showed low CSPs, making it unlikely that ParB$_{1-20}$ associates with alternative binding sites on the ParA dimer (Fig. 3b). Together, the mutational and NMR data strongly support the computational model of the ParA-ParB$_{1-20}$ complex.

## ParB- and DNA-binding induce structural changes in the Walker A and B regions of ParA

After identifying the ParB-binding site of ParA, we aimed to clarify the mechanism underlying the cooperative stimulatory effect of ParB- and DNA-binding on the ParA ATPase activity. The considerable distances of the bound ParB and DNA molecules from the catalytic site of the

ParA dimer suggested that they may not be directly involved in the mechanism of nucleotide hydrolysis, but rather induce conformational rearrangements in ParA that facilitate this process. To test this hypothesis, we set out to analyze the impact of these two ligands on the conformational state of ParA using hydrogen-deuterium exchange (HDX) mass spectrometry, a technique that detects local changes in the accessibility of backbone amide hydrogen atoms caused by conformational changes or ligand binding[73]. First, we determined the effect of ParB$_{1-20}$ on wild-type ParA•ATP dimers. The results showed that peptide binding led to a strong increase in HDX throughout the entire ParA dimer interface (Fig. 4a and Supplementary Fig. 9a). Thus, consistent with its limited but significant effect on the ParA ATPase activity (see Fig. 2a, b), the association of ParB$_{1-20}$ was sufficient to induce dissociation of the ParA dimer. This finding is in line with results from nucleotide content analyses showing that ParA•ATP dimers released the bound nucleotide upon prolonged incubation with ParB$_{1-20}$, whereas they stably retained it in the absence of the peptide (Fig. 4b). An even larger increase in HDX was observed in the presence of both the peptide and DNA, corroborating the synergistic effect of these two ligands (Supplementary Fig. 9a). By contrast, no dissociation or nucleotide release were observed when the same experiments were performed with the ATPase-deficient ParA-D60A variant (Supplementary Fig. 9b), supporting the notion that ParB$_{1-20}$ promotes ParA dissociation indirectly by stimulating ATP hydrolysis.

The high stability of the ParA-D60A dimer made it possible to analyze the conformational changes induced by ParB$_{1-20}$ and DNA binding that precede ATP hydrolysis and thus likely trigger this event. To this end, we compared the HDX patterns of ParA-D60A•ATP dimers in the absence or presence of different ligands. When the protein was incubated with DNA alone, we observed a strong reduction in HDX at the previously established DNA-binding interface of ParA[35,36] (Fig. 4c, d and Supplementary Fig. 9c), indicative of DNA binding. In this condition, there was no change in the HDX pattern at the catalytic site of ParA, consistent with the inability of DNA to induce appreciable ATP turnover in the absence of ParB (Figs. 2b and 4b). Unlike DNA, closed ParB clamps (ParB*) did not lead to a reduction of HDX on the surface of the ParA-D60A•ATP dimer (Fig. 4c, d and Supplementary Fig. 9c). This result indicates that ParB binding is relatively weak and transient, thereby not affecting the fast HDX reactions typically observed for surface-exposed protein regions[73]. However, there was a noticeable increase in HDX in a region at the catalytic site of ParA comprising the Walker B motif and an adjacent loop that reaches up to the ParB-binding site and contains the essential ParB-interacting residues L150 and L152 (Fig. 4c–e and Supplementary Fig. 9c). The association of ParB thus appears to induce a conformational change in this loop region that induces a rearrangement of the Walker B motif, thereby improving the efficiency of ATP hydrolysis (compare Fig. 2a). Importantly, when incubating ParA-D60A•ATP dimers with both closed ParB* clamps and DNA, we additionally observed an increase in the rate of HDX in the catalytic Walker A motif, which is linked to an α-helix adjacent to the DNA-binding pocket (Fig. 4c–e and Supplementary Fig. 9c). Very similar results were obtained when using the ParB$_{1-20}$ peptide instead of full-length ParB clamps (Supplementary Fig. 9b). Although the precise nature of the structural changes induced in the Walker A and Walker B motifs remains unclear, it is apparent that ParB and DNA act synergistically to remodel the catalytic site of ParA, explaining the cooperative stimulatory effects of these two ligands on the ParA ATPase activity.

## The arginine in the ParA-binding motif of ParB binds a conserved pocket on the ParA dimer surface

Previous studies have suggested that the central positively charged residue in the ParA-binding motif of ParB (R13 in *M. xanthus* ParB) might function as an arginine finger that reaches down to the catalytic site of ParA and directly contributes to the mechanism of ATP

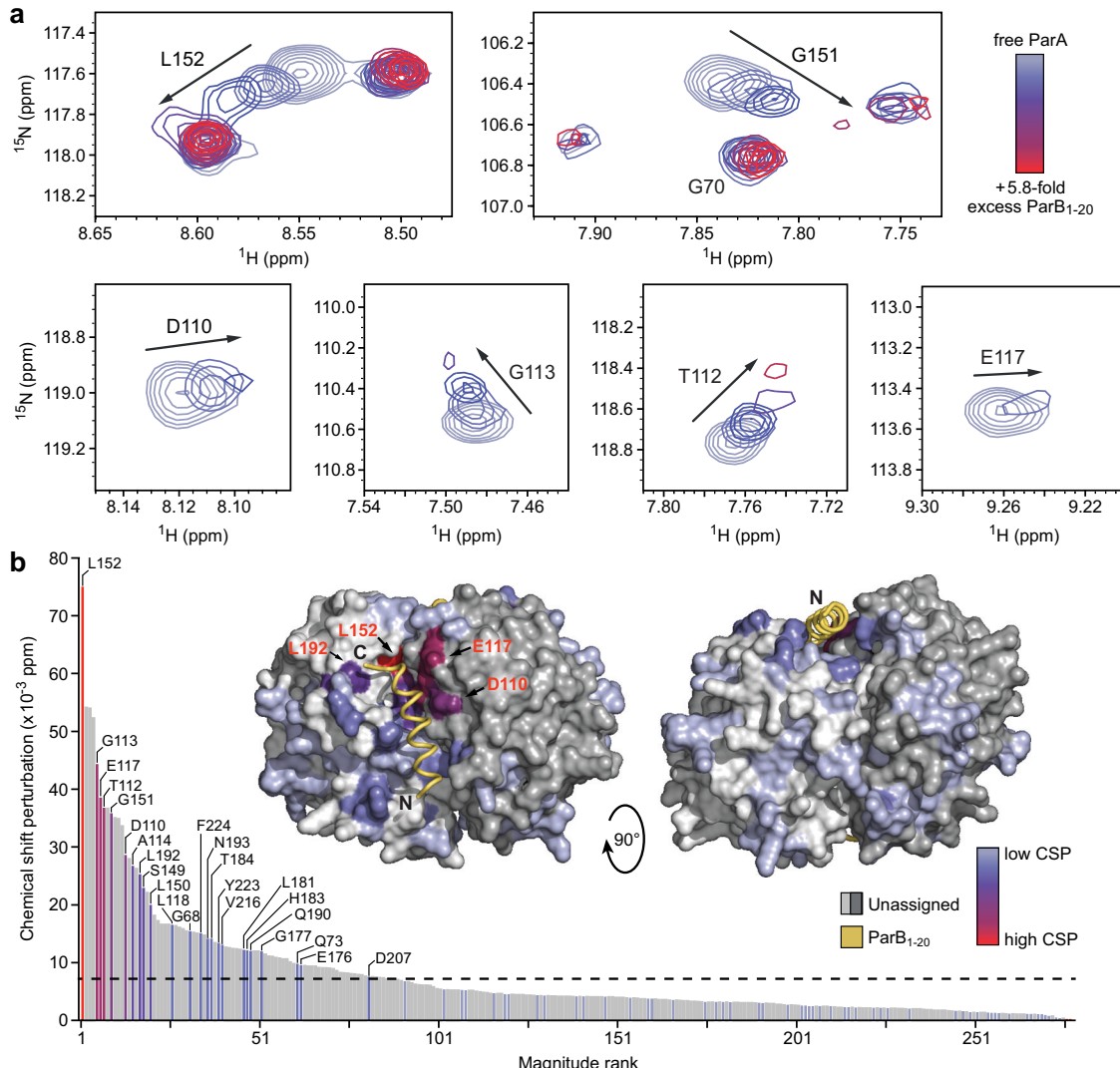

**Fig. 3 | NMR spectroscopy confirms the predicted ParB-binding site of ParA.**
**a** Chemical shift perturbations (CSPs) observed upon titration of isotopically labeled ParA$_{21-274}$·D60A dimers with increasing concentrations of unlabeled ParB$_{1-20}$ peptide, monitored using a two-dimensional $^1$H-$^{15}$N transverse relaxation-optimized heteronuclear single quantum correlation (TROSY-HSQC) experiment. Shown are overlaid spectra zooming into regions that have been assigned to specific residues and exhibit strong CSPs (full spectra available in Supplementary Fig. 8). **b** Ranking and assignment of the CSPs obtained in the HSQC experiment

described in (**a**). The CSPs observed in the presence of 0.7 molar equivalents of ParB$_{1-20}$ were sorted by magnitude, with assigned CSPs shown in a color gradient from light blue (lowest CSP) to red (highest CSP) and unassigned CSPs depicted in gray. The dashed line denotes the average CSP in the experiment. The inset shows a plot of the CSPs onto a structural model of the *M. xanthus* ParA$_{21-274}$ dimer in complex with two ParB$_{1-20}$ peptides (as in Fig. 2f), using the same color code as described above. The ParB$_{1-20}$ molecules are shown as yellow ribbons with their N- and C-termini indicated. The two ParA subunits are colored in shades of gray.

hydrolysis[65,66]. However, a large fraction (~50%) of chromosomally encoded ParB proteins carry a lysine at the corresponding position (Fig. 1c), which is not able to fulfill this function[74,75]. Apart from that, our structural model suggested that this positively charged residue is located opposite the hydrophobic face of the ParA-binding helix and thus not oriented towards the catalytic site of ParA (Fig. 2g). To clarify its role, we generated variants of *M. xanthus* ParB in which R13 was replaced with lysine or alanine, respectively, and then determined their ability to associate with ParA* using the BLI setup described above (see Fig. 1a). Interestingly, the mutant ParB proteins were still capable of recruiting ParA* dimers, with only a moderate reduction in their binding activity (Fig. 5a). However, their ability to stimulate the ParA ATPase activity was markedly reduced (R13K) or almost completely abolished (R13A) (Fig. 5b). To validate these results, we analyzed the functionality of the mutant ParB proteins in vivo (Fig. 5c–e). As expected, neither of the substitutions interfered with partition complex formation. The wild-type protein showed

robust chromosome segregation activity. Similarly, the R13K variant was still largely functional, although it gave rise to a small fraction of cells that displayed an abnormal number or distribution of partition complexes. The R13A variant, by contrast, was no longer able to support DNA segregation and produced highly abnormal segregation patterns similar to those obtained for the ParBΔ21 variant (compare Fig. 1e and Supplementary Fig. 3B). Collectively, these results show that the positively charged residue at the center of the ParA-binding motif critically contributes to ParB function, both by supporting the recruitment of ParA and mediating the stimulatory effect of ParB on its ATPase activity.

As the central positively charged residue of the ParA-binding motif of ParB is highly conserved, we hypothesized that it may interact with a conserved feature in ParA. To identify potential target sites, we inspected the structural model of the *M. xanthus* ParA$_{21-274}$ dimer in complex with ParB$_{1-20}$ (Fig. 2f) and identified the segment of ParA located adjacent to residue R13 of the bound peptide. Subsequently,

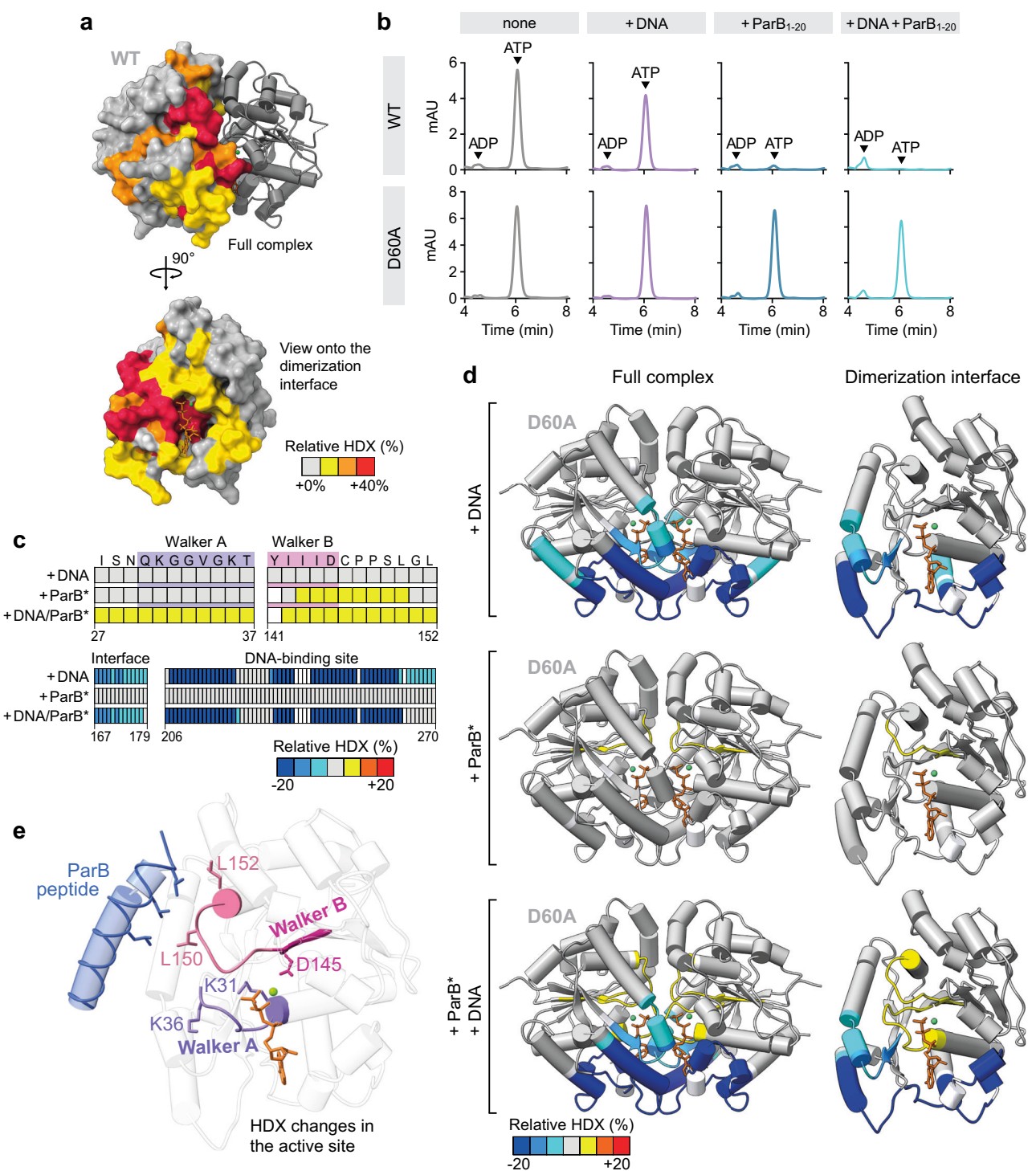

we used an amino acid sequence alignment of ParA homologs to search for conserved surface-exposed residues in this protein region (Fig. 6a, b). This analysis identified four invariant (Y81, E117) or functionally conserved (E82, D110) residues in ParA that are predicted to be located in the immediate vicinity of R13 (Fig. 6c) and, in part (D110, E117), also showed large CSPs upon ParB$_{1-20}$ binding in the NMR analysis (Fig. 3b). The aromatic ring of Y81 is predicted to be positioned close to the positively charged guanidine group of R13, enabling a cation-π interaction between these two amino acids[76]. This interaction may be stabilized by residues E82, D110 and E117, which form a negatively charged cluster surrounding Y81 that could establish electrostatic interactions with the R13 side chain (Fig. 6c).

To test whether the four residues identified could mediate the stimulatory effect of R13 on the ParA ATPase activity, we generated a mutant ParA variant in which Y81 was replaced by leucine (Y81L), a hydrophobic amino acid that is similar in length to tyrosine but unable to establish cation-π interactions. In addition, we generated variants in which the negatively charged amino acids E82, D110, and E117 were replaced, either individually (E82R, D110R, E117R) or pairwise (D110R/E117R), with a positively charged arginine residue. The mutant proteins were then analyzed for their ability to interact with model partition complexes using BLI (see Fig. 1a). The results showed that the different substitutions had only mild effects on the ParB-binding activity of ParA, with the Y81L variant still reaching wild-type binding levels and

**Fig. 4 | Cooperative ParB and DNA binding induces structural changes at the catalytic site of ParA. a** Hydrogen-deuterium exchange (HDX) analysis of the effect of ParB$_{1-20}$ on wild-type ParA dimers. Shown is the difference in deuterium uptake by ParA (50 µM) in deuterated buffer containing ATP (1 mM) in the presence and absence of ParB$_{1-20}$ (1 mM), mapped onto the crystal structure of the ParA$_{21-274}$-D60A•ATP dimer in surface representation (*t* = 1000 s; Supplementary Fig. 9a and Supplementary Data 1). **b** Nucleotide content analysis investigating the effect of DNA and ParB$_{1-20}$ on the ATPase activity of ParA under single-turnover conditions. Wild-type ParA or ParA-D60A (50 µM) was incubated with salmon sperm DNA (1 mg/mL) and/or ParB$_{1-20}$ (1 mM) for 30 min at room temperature. After denaturation of the proteins, the released nucleotides were separated by HPLC and detected at a wavelength of 260 nm. The data show a representative experiment (*n* = 2). **c** HDX analysis investigating the structural changes in ParA-D60A dimers induced by ParB and DNA binding. ParA-D60A (50 µM) was incubated in deuterated buffer containing ATP and CTP (1 mM each) either alone or in the presence of salmon sperm DNA (1 mg/mL) and/or ParB-Q52A (ParB*) (100 µM) pre-incubated with a *parS*-containing DNA stem-loop (2.5 µM) to induce its transition to the closed state. The

heatmap shows the maximal differences in deuterium uptake by ParA-D60A in the DNA-bound, ParB*-bound and DNA/ParB*-bound states compared to the apo-state for selected residues in the Walker A and Walker B regions, in a region at the dimer interface linking the DNA-binding site and the Walker A loop, and in the DNA-binding region (see Supplementary Fig. 9c and Supplementary Data 1 for details). (**d**) Global changes in the HDX pattern of ParA-D60A dimers upon incubation with DNA and/or ParB*. Shown are the maximal differences in deuterium uptake for each of the comparisons described in (**c**) mapped onto the crystal structure of the ParA$_{21-274}$-D60A•ATP dimer in cartoon representation. ATP molecules are shown in orange, Mg$^{2+}$ ions in green (see Supplementary Fig. 9c and Supplementary Data 1 for details). **e** Crystal structure of ParA$_{21-274}$-D60A•ATP with a modeled ParB$_{1-20}$ peptide taken from the predicted structure in Fig. 2f, shown in cartoon representation. ATP is depicted in orange, the Mg$^{2+}$ ion in green. Hydrophobic residues in ParB$_{1-20}$ (blue) and the Walker B-proximal loop of ParA (light red) are shown in stick representation. The Walker A (purple) and Walker B (magenta) motifs are highlighted, with catalytically relevant residues shown as sticks. Source data are provided as a Source data file.

all other variants showing only a moderate decrease in their affinity for ParB (Fig. 6d). Thus, the four residues do not appear to be critical for the association of the two proteins, reminiscent of the situation seen for residue R13 of ParB (compare Fig. 5a). However, while still binding to ParB and having (close-to-) normal DNA-binding activity (Supplementary Fig. 10), all mutant ParA variants showed a drastic reduction in their ability to undergo ParB-stimulated ATP hydrolysis (Fig. 6e). Interestingly, they almost (E82A) or completely (Y81L, D110R, E117R, D110R/E117R) lacked ATPase activity in reactions containing only ParB* and a limited amount of *parS* DNA, which was added to trigger ParB* clamp closure but also served as a source of non-specific DNA for ParA. By contrast, ATP turnover was clearly detectable for all proteins when the reactions were additionally supplemented with an elevated concentration (100 µg/mL; compare Fig. 2b) of salmon sperm DNA (Fig. 6e). However, even at saturating ParB concentrations, the turnover numbers reached remained markedly lower than those observed for the wild-type protein (Supplementary Fig. 11). Together, these findings suggest that the different amino acid substitutions not only impair the ParB-dependent stimulation of ATP hydrolysis but also affect the synergistic stimulatory effect of non-specific DNA on this process. Notably, the E110R/E117R variant of ParA displayed significant ATP turnover when incubated with non-specific DNA alone, whereas its ATPase activity was completely abolished in the presence of closed ParB* clamps (Fig. 6e). Thus, the introduction of two positive charges in the vicinity of Y81 may mimic the interaction with residue R13 of ParB but cause complications due to charge repulsion in the actual event of ParB binding. An inspection of the ParA crystal structure revealed that residues Y81 and E82 are located in an α-helix immediately adjacent to Motif 2, a region that is conserved in members of the ParA/MinD family[31,77–81] and includes three residues (D58, D60, N64) coordinating the Mg$^{2+}$ ion and the nucleophilic water at the catalytic site of ParA[31–34] (Fig. 6f). Residue R13 of ParB is therefore likely to promote conformational changes in the Motif 2 region that facilitate the transition of ParA to the hydrolysis-competent state in the presence of DNA.

## Discussion

The ParAB*S* system is highly conserved and critical for proper DNA segregation in the majority of bacteria[4]. Over the past three decades, considerable progress has been made in understanding its function, but the precise mode of interaction between ParA and ParB and the mechanism underlying the stimulatory effect of ParB on the ParA ATPase activity have remained unclear. In this work, we clarify these key functional aspects by comprehensively analyzing the functional interplay between ParA and ParB in the *M. xanthus* system. Our results show that the hydrophobic residues of the N-terminal ParA-binding motif of ParB interact with a conserved hydrophobic pocket at the

ParA dimer interface, whereas the single positively charged residue in this motif associates with a neighboring negatively charged surface region. Both binding sites are adjacent and directly connected to critical catalytic residues at the nucleotide-binding site of ParA. In line with this finding, we demonstrate that ParB and DNA binding act synergistically to alter the structure and/or conformational dynamics of the Walker A and Walker B regions in the ParA dimer, consistent with their cooperative effects on the ParA ATPase activity. Importantly, ParA exhibits a strong increase in DNA-binding affinity when bound to ParB and interacts preferentially with closed ParB clamps (Fig. 7a). This behavior minimizes the futile activation of the ParA ATPase activity by freely diffusible, open ParB clamps. Moreover, it ensures that ParA dimers become locked in place on the nucleoid once they are in contact with a partition complex, which allows them to withstand the considerable elastic forces exerted during the segregation process[42]. The lower DNA-binding affinity in the absence of ParB, on the other hand, may facilitate their diffusive motion on the nucleoid and thus ensure the steady differential flux of ParA dimers towards sister partition complexes required to drive directional DNA segregation[48,82]. Our findings shed light on the mechanism of ParAB*S*-mediated DNA partitioning and thus close critical gaps in the understanding of bacterial chromosome and plasmid segregation. Given the high functional and structural conservation of ParA-like ATPases, they may also help to clarify the modes of action of analogous segregation systems that mediate the position of macromolecular cargoes such as chemoreceptors and carboxysomes[21].

### Mode of interaction between ParA and ParB in *M. xanthus*

We show that *M. xanthus* ParA binds a conserved peptide in the N-terminal region of ParB that contains a series of hydrophobic residues interrupted by a central positively charged residue, as described previously for other ParAB*S* systems[31,64]. This interaction is largely dependent on the hydrophobic residues of the ParA-binding peptide, consistent with results from mutant screens and two-hybrid analyses suggesting that these residues are critical for a functional interaction between ParA and ParB in *B. subtilis*[83] and *P. aeruginosa*[64], respectively. A combination of computational modeling, NMR analysis and mutational studies revealed that these key interaction determinants insert into a hydrophobic cleft at the edge of the ParA dimer interface, with the majority of direct interactions limited to one of the two ParA subunits. The central positively charged residue (R13) of the ParA-binding peptide, by contrast, associates with a conserved site adjacent to this cleft that is formed by surface-exposed residues in helices H4 and H6/H7 of the *trans*-subunit, in a process likely mediated by electrostatic interactions with three acidic residues and a cation-π interaction with an invariant surface-exposed tyrosine residue. Such cation-π interactions are widespread in proteins, with arginine-tyrosine pairs

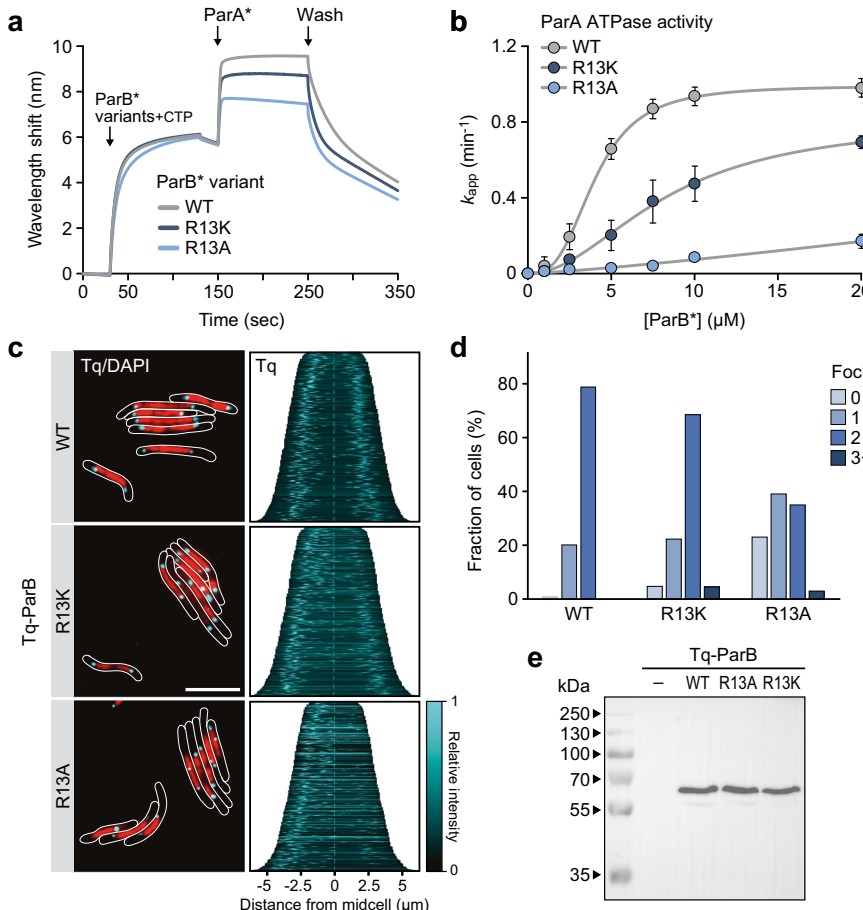

**Fig. 5 | Residue R13 of ParB is dispensable for ParA binding but critical for ATPase stimulation. a** BLI analysis of the interaction of ParA with DNA-bound ParB-Q52A (ParB*) variants lacking the conserved residue R13. The indicated ParB-Q52A variants were loaded onto a closed *parS*-containing DNA fragment (see Fig. 1a) and probed with ParA-R238E (ParA*) (5 μM) in the presence of ATP and CTP (1 mM each). At the end of the association phase, the biosensors were transferred into nucleotide- and protein-free buffer to monitor the dissociation reactions. Shown are the results of a representative experiment ($n = 3$ independent replicates). **b** Stimulation of the ParA ATPase activity by ParB-Q52A (ParB*) variants lacking R13. Shown is the ATPase activity of ParA (5 μM) in the presence of increasing concentrations of the indicated ParB-Q52A variants in reactions containing ATP (1 mM), CTP (1 mM), salmon sperm DNA (100 μg/mL) and a *parS*-containing DNA stem-loop (250 nM). Data represent the mean of four independent replicates (±SD), normalized to reactions without ParB*. The results were fitted to a Hill equation. **c** Subcellular localization of ParB variants lacking R13 in *M. xanthus*. Cells producing Tq-ParB (MO072), or Tq-ParB-R13K (LS005) or Tq-ParB-R13A (LS004) in place of wild-type ParB were stained with DAPI prior to analysis by phase contrast and fluorescence microscopy. The images show overlays of the Tq and DAPI signals, with the cell outlines indicated in white (bar: 5 μm). The demographs on the right summarize the subcellular distribution of the Tq signal in representative subpopulations of cells ($n = 400$ per strain). **d** Quantification of the number of fluorescent foci in cells producing wild-type (WT) Tq-ParB (MO072, $n = 358$) or its R13K (LS005, $n = 422$) or R13A (LS004, $n = 369$) derivative in place of the native ParB protein. **e** Immunoblot analysis of the strains analyzed in panel c with anti-GFP antibodies. A ΔparB mutant producing untagged ParB under the control of an inducible promoter (SA4269) was used as a negative control (−). Shown is a representative image ($n = 3$ independent replicates). Source data are provided as a Source data file.

representing the most common combination of interacting residues[84]. The fact that the ParB-binding pocket is constituted by amino acids from both ParA subunits explains why the association of ParB can only occur upon ParA dimerization. Importantly, the location of the ParB binding site established in this study is consistent with results obtained for other ParA orthologs (Supplementary Fig. 12). For instance, chemical crosslinking studies of the ParA homolog of plasmid pSM19035 showed that the region stretching from helix H4 to H6 is located in close proximity to the bound ParB protein[68]. Moreover, amino acid substitutions in the vicinity of helix H4 were found to markedly reduce the ParB-binding activity of *P. aeruginosa* ParA in vivo[67]. A different mode of interaction has recently been proposed for the ParA homolog of *H. pylori*, based on a crystal structure of the protein in complex with a short N-terminal ParB peptide generated spontaneously during crystal growth[69]. However, in this structure, the peptide only associates with a single ParA chain, without any contact to the *trans*-subunit, which is difficult to reconcile with the dependence of ParB binding on ParA dimerization. Moreover, it lacks a large part of the conserved ParA-interaction motif and is only loosely attached to the ParA surface, raising questions as to the specificity of the interaction observed. Yet another binding mode was suggested by crystallographic studies of the ParA homolog of plasmid TP228[66]. In this case, N-terminal peptides derived from the cognate centromere-binding protein ParG were found to associate with a region of ParF that corresponds to helices H10 and H11 of *M. xanthus* ParA, with only few interactions to the respective *trans*-subunits. The binding site obtained still remains to be verified by other experimental approaches. However, given the significant evolutionary distance of ParF from the aforementioned chromosomal and related plasmid-encoded ParA orthologs[85], it is possible that this protein has evolved a distinct mode of interaction with its ATPase-activating peptide.

## Stimulatory effect of ParB on the ParA ATPase activity
Our results indicate that ParB uses a dual mechanism to stimulate the ParA ATPase activity. On the one hand, the hydrophobic residues of the ParA-binding motif interact with hydrophobic residues in a

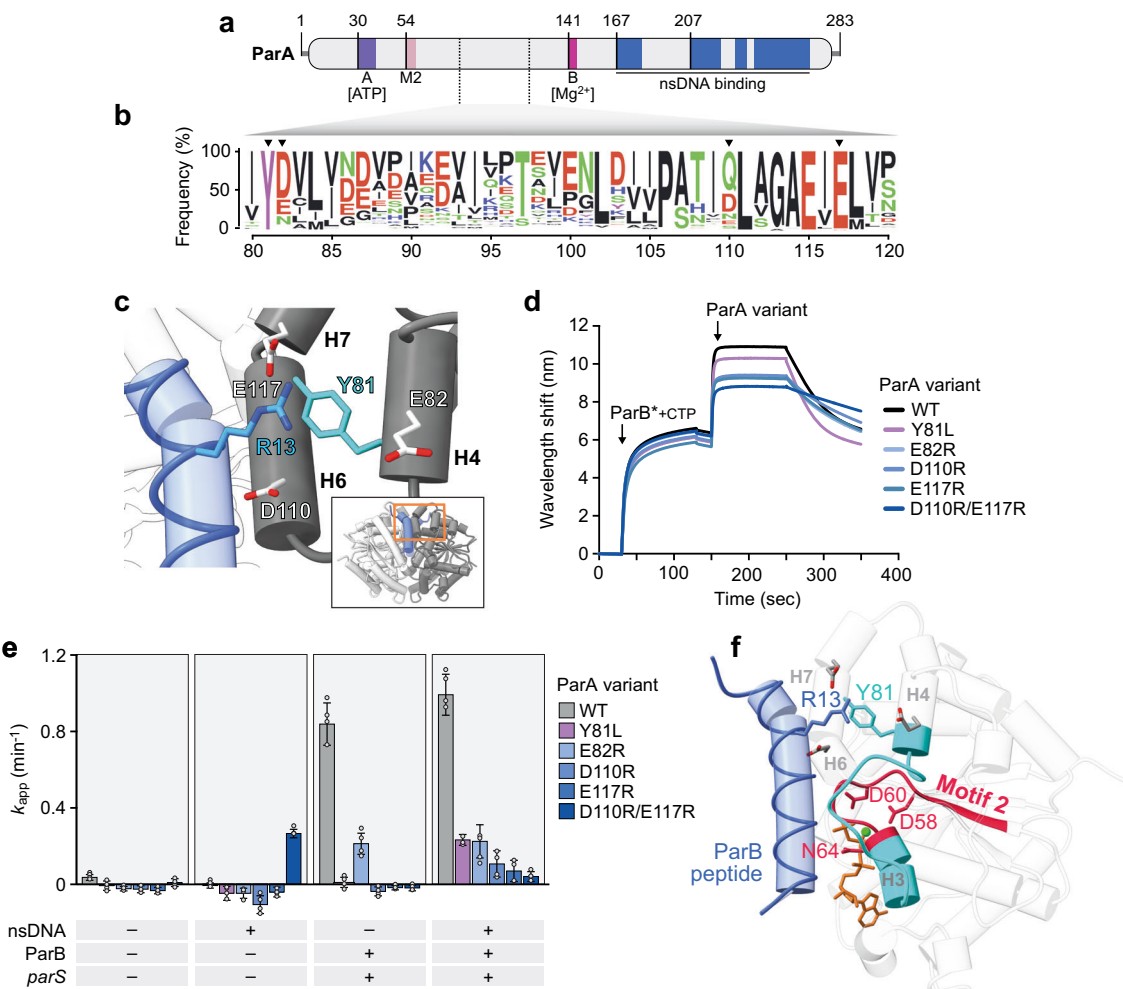

**Fig. 6 | Residue R13 of ParB binds a conserved site adjacent to the ParA dimer interface. a** Schematic representation of *M. xanthus* ParA. Regions involved in ATP binding and hydrolysis, including the Walker A motif (purple), Motif 2 (light red), and the Walker B motif (magenta), are highlighted. Regions that show DNA-dependent protection in HDX experiments (see Fig. 4c, d) are shown in blue. **b** Conservation of the helix H4/H6/H7 region of ParA. The graph shows a sequence logo of the helix H4/H6/H7 region based on an alignment of 3800 ParA homologs obtained by protein BLAST analysis with *M. xanthus* ParA as a query. Residues are colored according to their physico-chemical properties (as in Fig. 1c). The numbering indicates the corresponding residues in *M. xanthus* ParA. Residues investigated in this study are highlighted by arrowheads. **c** Magnification of the predicted binding site of residue R13 of ParB on the ParA dimer surface, taken from the model in Fig. 2f. Interacting residues are shown in stick representation and labeled, with R13 modeled in one of its possible rotameric states. The inset indicates the location of the magnified region in the ternary complex (orange rectangle). **d** BLI analysis of the interaction between DNA-bound ParB and ParA variants with amino acid

exchanges in the helix H4/H6/H7 region. ParB-Q52A (ParB*) (10 μM) was loaded onto a closed *parS*-containing DNA fragment (see Fig. 1a) and probed with the indicated ParA variants (5 μM) in the presence of ATP (1 mM) and 500 mM KCl. At the end of the association phase, the biosensor was transferred into a protein- and nucleotide-free buffer to follow the dissociation kinetics. **e** Stimulatory effect of ParB on ParA variants with amino acid substitutions in the H4/H6/H7 region. Shown are the ATPase activities of the indicated ParA variants (2.5 μM) in the presence of ATP and CTP (1 mM each) with or without ParB-Q52A (10 μM), salmon sperm DNA (100 μg/mL), and/or a *parS*-containing DNA stem-loop (250 nM). Data represent the mean of four independent replicates (±SD). **f** Crystal structure of ParA$_{21-274}$-D60A•ATP with a modeled ParB$_{1-20}$ peptide taken from the predicted structure in Fig. 2f, shown in cartoon representation. The ParB$_{1-20}$ peptide (blue), Motif 2 with its catalytic residues D58, D60, and N64 (dark red), and the region connecting Motif 2 and helix H4 (cyan) are highlighted. ATP is depicted in orange, the Mg$^{2+}$ ion in green. Relevant residues are displayed in stick representation, with R13 shown in one of its possible rotameric states. Source data are provided as a Source data file.

surface-exposed loop of the *cis*-subunit that is directly connected to the Walker B motif of ParA. On the other hand, the central positively charged residue R13 likely associates with the helix H4/H6/H7 region of the *trans*-subunit, which is immediately adjacent to the conserved Motif 2. These two motifs contain residues that coordinate the triphosphate moiety of ATP, the Mg$^{2+}$ ion and catalytic water molecules at the active site and thus play a critical role in the mechanism of nucleotide hydrolysis[31,77,79–81]. HDX analysis showed that ParB binding leads to an increased accessibility of the Walker B region, indicating a structural rearrangement at the catalytic site of ParA. The lack of obvious changes in HDX around Motif 2 may be explained by the fact

that the corresponding region is tightly embedded in the protein and thus less accessible to the environment[31], which strongly reduces the HDX rate[73]. Alternatively, the structural changes induced may be too subtle to be detected by this technique. An additional region of elevated HDX was observed if ParA was associated with both ParB and non-specific DNA, including the deviant Walker A motif, which coordinates the phosphate moieties of ATP and thus also critically contributes to the catalytic mechanism of ParA[31,77–81]. Given the considerable distance between the Walker A motif and the DNA binding site, this effect may reflect a rearrangement in the global structure of ParA that is induced cooperatively by ParB and DNA

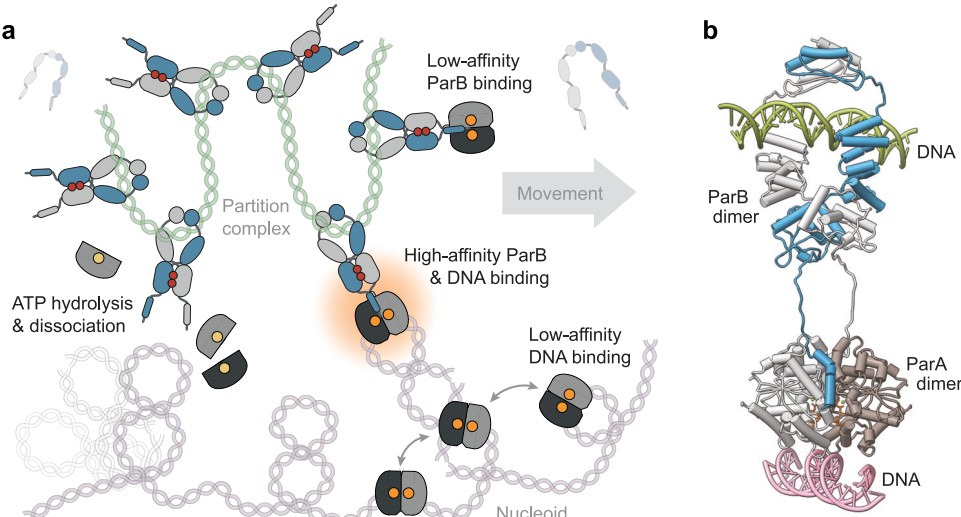

**Fig. 7 | Model of the ParA-ParB interaction. a** Role of the ParA-ParB interaction in DNA segregation. ParB clamps are loaded onto the centromere region and form a densely packed partition complex. ParA dimers, by contrast, diffuse across the nucleoid in a low-DNA-affinity state, which ensures a steady flux of dimers towards the partition complex. Closed ParB clamps interact with diffusible DNA-bound ParA dimers, forming a tethering complex that links the partition complex to the nucleoid. Cooperative interactions between the ParB- and DNA-binding sites stabilize the tethering complex and thus enables it to harness the elastic dynamics of chromosomal DNA loops for partition complex movement. The transition to this locked state involves structural rearrangements at the catalytic site of ParA that stimulate its ATPase activity, thereby limiting the lifetime of the tethers. ATP

hydrolysis then leads to the dissociation of ParA and its release from both ParB and DNA, allowing the handover of the partition complex to adjacent DNA-bound ParA dimers. **b** Structural model of the *M. xanthus* ParA-ParB tethering complex. Shown is a model of a ParB dimer loaded onto *parS*-containing DNA (green) and interacting with a DNA (pink)-bound ParA dimer. ATP is shown in orange, $Mg^{2+}$ in green. The ParB$_2$-*parS* and ParA$_2$-(ParB$_{1-20}$)$_2$ complexes were modeled separately with Alpha-Fold 3[126] and then joined using UCSF-Chimera[103]. The DNA molecule bound to the ParA dimer was fitted into the model based on a superimposition of the predicted ParA$_2$-(ParB$_{1-20}$)$_2$ complex with the crystal structure of a DNA-bound *H. pylori* ParA-D41A•ADP dimer (PDB: 6IUD[35]).

binding. In support of this hypothesis, the association of ParB strongly increases the affinity of ParA for non-specific DNA, indicating a structural coupling of the two binding sites. However, the increased accessibility of the Walker A motif observed by HDX analysis could also be due to a slight shift in the positioning of two ATP molecules, mediated by direct interactions of their nucleobases with α-helices and loops that are immediately adjacent to the bound DNA molecule[35,36]. Importantly, the structural changes observed closely correlate with the cooperative effect of ParB and DNA on the ParA ATPase activity, supporting a causal link between these two phenomena. Collectively, our results indicate that ParB stimulates the ATPase activity of ParA by inducing structural changes in the vicinity of its binding site that modulate the architecture of ParA's catalytic center. They also suggest that the central positively charged residue (R13) in the ParA-binding motif of ParB does not act as an arginine finger, as proposed previously[65,66], but rather facilitates this structural transition by interacting with the helix H4/H6/H7 region. This notion is supported by the fact that many ParB proteins contain a lysine instead of an arginine in their N-terminal ParA-binding motif (Fig. 1e), which is typically unable to functionally replace an arginine finger[74,75]. Accordingly, the substitution of R13 with lysine has only a moderate effect on the function of *M. xanthus* ParB (Fig. 5b–d). Moreover, for a ParB peptide bound at the dimer interface, the distance of the central arginine residue from the triphosphate moiety of ATP is too large to permit a direct involvement in the catalytic reaction. Apart from promoting structural changes in ParA, R13 may also be critical to ensure the proper folding and positioning of the ParA-binding peptide upon contact with its binding pocket. As the only polar residue in the ParA-binding motif, its exposure to the solvent may lock the peptide in an orientation that facilitates its transition to the α-helical state, driven by Van der Waals interactions of the neighboring residues with the hydrophobic binding pocket. This process may be further facilitated by the association of R13 with the helix H4/H6/H7 region of ParA, which holds the peptide in place

and thus ensures the correct positioning of its hydrophobic residues in the binding interface.

The ParA dimer contains two ParB-binding sites located face-to-face on opposite sides of the dimer interface. Therefore, the association of two ParB peptides may be required to fully activate the ATPase activity of ParA. In line with results obtained for the ParA homolog of the *E. coli* F plasmid[19], we observed that full-length ParB dimers have a considerably higher stimulatory activity than synthetic peptides containing a single ParA-binding motif, likely because the linkage of the two N-terminal peptides in the dimeric complex strongly increases the avidity of the interaction and thus ensures the simultaneous occupancy of both ParB-binding sites. The stimulatory effect of ParB dimers increases further upon the CTP-dependent closure of their DNA entry gate, which juxtaposes the two N-terminal peptides and thus increases their local concentration. This mechanism ensures that ParA only shows significant ATPase activity when it has associated with both DNA and ParB, thereby preventing the premature release of ParA dimers from the nucleoid and limiting the lifetime of the ParA-mediated tethers between chromosomal loops and partition complexes[42] once the complex has formed. Cooperative interactions between the two ParB-binding sites of a ParA dimer could potentially be mediated by a conserved glutamine residue (Q62) in Motif 2, which forms hydrogen bonds with the backbone of both the Walker B-proximal loop and the P-loop in the *trans*-subunit (Supplementary Fig. 13). By connecting these critical catalytic motifs, it may be able to relay structural changes between the catalytic centers and the adjacent ParB-binding sites across the dimer interface. However, it was not possible to test this hypothesis because a ParA variant lacking this residue proved to be insoluble.

### Conservation of the ParB-binding site among ParA homologs
Similar to ParA, most other members of the ParA/MinD family reported to date interact with factors that stimulate their ATPase activity[21]. In many species, two or more of these proteins coexist without interfering with each other's function, indicating a high

degree of specificity in the interactions involved. A comparison of known crystal structures shows that the structural elements constituting the ParB site of *M. xanthus* ParA are conserved in a variety of different ParA orthologs (Supplementary Figs. 14a–d). In the chromosomally encoded ParA proteins of *M. xanthus*, *H. pylori*[35] and *Thermus thermophilus*[31], the folds and the amino acid sequences of both the Walker B-proximal loop and the helix H4/H6/H7 region are very similar. Accordingly, the N-terminal regions of the corresponding ParB proteins all contain similar ParA-binding motifs (Supplementary Fig. 14d). Given the large evolutionary distance of the three species, this observation suggests that the mode of interaction between ParA and ParB identified in the present study may be widely conserved among chromosomal ParAB*S* systems. Notably, the Walker B-proximal loop and the helix H4/H6/H7 region are also conserved in various plasmid-encoded ParA orthologs, including those of *Vibrio cholerae* chromosome 2[86] and plasmid pSM19035[80] (Supplementary Fig. 12a–d). However, their cognate ParB proteins do not feature the typical ParA-binding motif of their chromosomally encoded counterparts (Supplementary Fig. 14d). This finding suggests that, in these systems, the ParA-ParB interaction may still be mediated by the same protein regions but rely on different interaction determinants to prevent crosstalk with the chromosomal systems of the host cells. More distantly related ParA orthologs, by contrast, including those encoded by plasmid TP228[66], plasmid pNOB8[87] or the chromosome of *Sulfolobus solfataricus*[88], show significant differences in both the architecture of the helix H4/H6/H7 region and the sequence of the ATPase-stimulating peptide, potentially indicating a different mode of ParB binding.

Notably, the determinants constituting the ParB-binding pocket of ParA are largely missing in MinD from *E. coli*[89] (Supplementary Figs. 14 and 15), another prototypical member of the ParA/MinD family, whose function depends on the stimulation of its ATPase activity by the topological specificity factor MinE[22]. Previous work has shown that the N-terminal region of MinE associates with a different site at the MinD dimer interface, formed by the C-terminal regions the two subunits (Supplementary Fig. 15). In the resulting complex, the conserved positively charged residue R21 of MinE interacts with the negatively charged residue E53 of MinD, which is not conserved in ParA orthologs and exposed at the edge of the dimer interface in close proximity to Motif 2[89]. Notably, it has been proposed that the stimulatory effect of MinE relies on structural rearrangements in this catalytic motif that remodel the active site of MinD and thus increase the efficiency of the hydrolytic reaction[90]. The binding of regulatory proteins to the Motif 2 region may therefore represent a conserved mechanism to trigger ATP hydrolysis in this group of proteins, although it may be achieved through interaction with different, system-specific binding sites.

## Methods

### Experimental models
The bacterial strains used in this study are derivatives of *Myxococcus xanthus* DK1622[91], *Escherichia coli* TOP10 (Thermo Fisher Scientific), or *Escherichia coli* Rosetta(DE3)pLysS (Merck). The amino acid sequences of ParA (UniProt: Q1CVJ3) and ParB (UniProt: Q1CVJ4) correspond to the translation products of ORFs MXAN_7477 and MXAN_7476 in the genome sequence of *M. xanthus* DK1622[92], respectively.

### Cultivation of bacterial strains
*M. xanthus* DK1622 and its derivatives were grown at 32 °C in CTT medium[93]. If necessary, media were supplemented with kanamycin (50 µg/mL) or oxytetracycline (10 µg/mL). The expression of genes placed under the control of the the $P_{cuo}$[94], $P_{van}$ or $P_{lac}$[95] promoters was induced by supplementation of the media with 300 µM $CuSO_4$, 500 µM sodium vanillate or 1 mM isopropyl-β-D-thiogalactopyranoside (IPTG), respectively. *E. coli* strains were grown at 37 °C in LB medium

supplemented with ampicillin (200 µg/mL), kanamycin (liquid: 30 µg/mL, solid: 50 µg/mL), or chloramphenicol (liquid: 20 µg/mL, solid: 30 µg/mL), unless indicated otherwise.

### Plasmid and strain construction
The construction of bacterial strains and plasmids is described in Supplementary Tables 2 and 3, respectively. The oligonucleotides used are listed in Supplementary Table 4. Plasmids were propagated in *E. coli* TOP10 (Invitrogen) and verified by DNA sequencing (Microsynth Seqlab). *M. xanthus* was transformed by electroporation[96]. Non-replicating plasmids were inserted into the *M. xanthus* chromosome by site-specific recombination at the phage Mx8 *attB* site[97]. Their proper integration was confirmed by colony PCR.

### Protein purification
To purify hexahistidine-tagged **ParA** (His$_6$-ParA), *E. coli* Rosetta(DE3) pLysS cells carrying plasmid pMO105 were grown at 37 °C in 3 L of LB medium supplemented with ampicillin. At an optical density at 600 nm (OD$_{600}$) of 0.6, the temperature was decreased to 18 °C and His$_6$-ParA production was induced by the addition of IPTG to a final concentration of 1 mM. After incubation of the cultures overnight at 18 °C, and cells were harvested by centrifugation for 15 min at 10,000 × $g$ and 4 °C. The pellet was washed with buffer ParA1 (100 mM HEPES/KOH pH 7.4, 100 mM KCl, 100 mM L-arginine hydrochloride, 100 mM potassium L-glutamate, 5 mM $MgCl_2$, 1 mM EDTA, 10% [v/v] glycerol) and resuspended in 25 mL of buffer ParA1 supplemented with 10 µg/mL DNase I, 100 µg/mL phenylmethylsulfonyl fluoride and 1 mM β-mercaptoethanol. Subsequently, KCl (from a 4 M stock solution) was added to a final concentration of 1 M. The cells were disrupted by two passages through an LM20 Microfluidizer® (Microfluidics) at 16,000 psi, and cell debris was removed by centrifugation for 30 min at 30,000 × $g$ and 18 °C. The supernatant was cleared by syringe filtration and applied onto a 5 mL HisTrap HP column (GE Healthcare) equilibrated with buffer ParA2 (100 mM HEPES/KOH pH 7.4, 50 mM L-arginine hydrochloride, 50 mM potassium L-glutamate, 5 mM $MgCl_2$, 1 mM dithiothreitol [DTT], 10% [v/v] glycerol) containing 40 mM imidazole. After washing of the column with 50 mL of buffer ParA2, protein was eluted with a linear imidazole gradient (40–300 mM in buffer ParA2) at a flow rate of 2 mL/min. Fractions containing high concentrations of His$_6$-ParA were identified by SDS-PAGE, pooled, and applied onto a 5 mL HiTrap Q HP column (Cytvia) equilibrated with buffer ParA3 (100 mM HEPES/KOH pH 7.4, 50 mM L-arginine hydrochloride, 50 mM potassium L-glutamate, 5 mM $MgCl_2$, 1 mM DTT, 10% [v/v] glycerol). The column was washed with 50 mL of buffer ParA3, and protein was eluted with a linear KCl gradient (0–1 M in buffer ParA3) at a flow rate of 2 ml/min. Fractions containing His$_6$-ParA in high concentrations and purity were pooled, snap-frozen, and stored at −80 °C until further use. Mutant His$_6$-ParA variants were purified in the same manner. To prepare $^2H/^{13}C/^{15}N$-labeled His$_6$-ParA$_{21-274}$-D60A for NMR studies, cells were cultivated in 2 L of $^2H$ $^{13}C$ $^{15}N$ High Performance OD2 Media solution for *E. coli* (Silantes; isotopic enrichment >98 atom%), and the protein was purified as described above.

To purify wild-type **ParB**, *E. coli* Rosetta(DE3)pLysS cells carrying plasmid pMO104 were cultivated at 37 °C in 3 L LB medium supplemented with ampicillin until they reached an OD$_{600}$ of 0.6. Subsequently, the temperature was decreased to 18 °C and IPTG was added to a final concentration of 1 mM to induce the overproduction of His$_6$-SUMO-ParB. After incubation of the cultures overnight at 18 °C, the cells were harvested by centrifugation for 15 min at 10,000 × $g$ and 4 °C. The pellet was washed with 30 mL of buffer ParB1 (25 mM HEPES/NaOH pH 7.6, 300 mM NaCl, 0.1 mM EDTA, 5 mM $MgCl_2$), and resuspended in 30 mL of buffer ParB2 (25 mM HEPES/NaOH pH 7.6, 1 M NaCl, 0.1 mM EDTA, 5 mM $MgCl_2$, 30 mM imidazole) supplemented with 10 µg/mL DNaseI, 100 µg/mL PMSF and 1 mM DTT. The cells were disrupted by two passages through an LM20 Microfluidizer® (Microfluidics) at 16,000 psi, and cell debris was removed by centrifugation

for 30 min at $30,000 \times g$ and 4 °C. The supernatant was cleared by syringe filtration and applied onto a 5 mL HisTrap HP column (GE Healthcare) equilibrated with buffer ParB2. After washing of the column with 50 mL of the same buffer, protein was eluted with a linear imidazole gradient (30–300 mM) obtained by mixing buffer ParB2 with buffer ParB3 (25 mM HEPES/NaOH pH 7.6, 150 mM NaCl, 0.1 mM EDTA, 5 mM MgCl$_2$) containing 300 mM imidazole at a flow rate of 2 mL/min. Fractions containing high concentrations of His$_6$-SUMO-ParB were identified by SDS-PAGE, pooled and dialyzed overnight at 4 °C against 3 L buffer of ParB3 supplemented with Ulp1 protease and DTT (1 mM) to cleave the His$_6$-SUMO tag[98]. The solution was cleared by syringe filtration and applied onto a 5 mL HisTrap HP column (GE Healthcare) equilibrated with buffer ParB3 containing 30 mM imidazole, followed by a wash with the same buffer. His$_6$-SUMO was retained on the column, whereas ParB was recovered in the wash fractions. Fractions containing high concentrations of ParB were pooled and applied onto a 5 mL HiTrap SP HP column (Cytviva) equilibrated with buffer ParB3. After washing of the column with 50 mL of buffer ParB3, protein was eluted with a linear NaCl gradient (0.15–1 M) obtained by mixing buffer ParB3 with a similar buffer containing 1 M NaCl at a flow rate of 2 mL/min. Fractions containing high concentrations of pure ParB were pooled, snap-frozen, and stored at −80 °C until further use. Mutant ParB variants were purified in the same manner.

## Crystallization and structural analysis of ParA

*M. xanthus* His$_6$-ParA$_{21-274}$-D60A (1 mM) was incubated with 10 mM ATP and 5 mM (f.c.) ParB$_{1-20}$ peptide for 60 min at room temperature and then subjected to crystallization screens, performed at 20 °C using the sitting-drop method with 250-nL drops that consisted of equal parts of protein and precipitation solution. Crystals were obtained with a precipitate solution containing 0.1 M 2-(N-morpholino)ethanesulfonic acid (MES) pH 6.0, 10% (v/v) glycerol, 30% (v/v) PEG-600, and 5% (v/v) PEG-1000. Data were collected under cryogenic conditions at the ID23-1 beamline of the European Synchrotron Radiation Facility (Grenoble, France), processed with XDS and scaled with XSCALE[99]. Phase-determination was achieved by molecular replacement with PHASER[100], using the structure of the ParA homolog Soj from *Thermus thermophilus* (PDB: 2BEK)[31] as a model. The structure of ParA was then built manually in COOT[101] and refined with PHENIX 1.19.1[102]. Structural data were visualized with UCSF ChimeraX 1.5[103].

## Analytical size-exclusion chromatography

ParA or ParA$_{D60A}$ were diluted to 2.5 mg/mL (75 μM) in SEC buffer (25 mM Tris/HCl, 200 mM KCl, 5 mM MgSO$_4$, 2 mM DTT, pH 7.4). When indicated, ParB$_{1-20}$ peptide was added at a 1:20 molar excess over ParA (1.2 mM), followed by incubation of the reactions at room temperature for 3 min prior to analysis. After centrifugation at $10,000 \times g$ for 5 min, the mixtures were loaded onto a Superdex 75 3.2/300 (Cytviva) size exclusion column equilibrated with SEC buffer. Proteins and released nucleotides were then separated at a constant flow rate of 0.05 mL/min over a period of ~1 h and detected in the eluate photometrically at 280 nm.

## Biolayer interferometry

BLI analyses were performed with a BLItz® system equipped with High Precision Streptavidin 2.0 (SAX2) Biosensors (Sartorius). All experiments were performed in BLItz binding buffer (25 mM HEPES/KOH pH 7.6, 100 mM KCl, 10 mM MgSO$_4$, 1 mM DTT, 10 mM bovine serum albumin, 0.01% [v/v] Tween 20), unless indicated otherwise. A DNA fragment (234 bp) containing a single central *parS* site was PCR-amplified from *M. xanthus* chromosomal DNA with primers carrying a 5′-biotin-triethylene glycol (TEG) group (BioTEG-*parS*-3-for/BioTEG-*parS*-3-rev)[12]. After purification with a GenElute™ PCR Clean-Up kit (Sigma, USA), the resulting double-biotinylated fragment (100 nM) was immobilized on a biosensor, and a stable baseline was established.

The DNA was then loaded with the indicated ParB proteins in the presence of 1 mM CTPγS (for wild-type ParB and its derivatives) or 1 mM CTP (for the hydrolysis-deficient ParB-Q52A variant and its derivatives). After the establishment of a stable second baseline in protein-free buffer, the association of the indicated ParA proteins was monitored in the presence of 1 mM ATP and 1 mM CTPγS or CTP (as specified above). At the end of the association step, the biosensor was transferred into protein- and nucleotide-free buffer to follow the dissociation kinetics. At the end of the reaction, the biosensor was treated with 500 mM EDTA to remove all proteins from the immobilized DNA fragment and then re-equilibrated in BLItz binding buffer prior to the next binding assay. Experiments investigating the DNA-binding behavior of ParA in the absence or presence of the ParB$_{1-20}$ peptide were conducted as described above, but without the ParB-loading step.

## Nucleotide hydrolysis assay

ATP hydrolysis was measured using a coupled enzyme assay linking nucleotide hydrolysis to the oxidation of NADH[104,105]. The reactions contained ParA (5 μM or 2.5 μM, as indicated), 1 mM ATP, 20 U/mL pyruvate kinase, 20 U/mL L-lactate dehydrogenase, 800 μg/mL NADH and 3 mM phosphoenolpyruvate in 200 μL ParA reaction buffer (25 mM HEPES/KOH pH 7.4, 100 mM KCl, 10 mM MgSO$_4$, 1 mM DTT). When appropriate, the mixtures additionally contained ParB$_{1-20}$ peptide or ParB-Q52A variants at the indicated concentrations as well as 100 μg/mL double-stranded salmon sperm DNA. Reactions containing "closed" ParB-Q52A variants were supplemented with 1 mM CTP and 250 nM of a *parS*-containing DNA stem-loop (54 bases; *parS*-Mxan-wt). After preincubation for 10 min at 30 °C, the reactions were started by the addition of ParA. Subsequently, 150 μL of the reaction mixtures were transferred into a 96-well microtiter plate and ATP hydrolysis was followed by measuring the decrease in NADH absorbance at 340 nm at 2-min intervals with an Epoch2 plate reader (BioTek). Reactions without ParA were used to correct the values obtained for spontaneous nucleotide hydrolysis and NADH oxidation. The apparent turnover numbers ($k_{app}$) were determined by linear regression analysis in Microsoft Excel 2019. The results of titration series were fitted to the modified Hill equation $k_{app} = (v_{max}*[L]^n)/(K_A^n + [L]^n)+v_0$, where $k_{app}$ is the apparent turnover number at the *total* ligand concentration [L], $v_{max}$ the maximal apparent turnover number reached, $n$ the Hill coefficient, $K_A$ the *total* ligand concentration for which $k_{app} = 0.5*v_{max}$, and $v_0$ the basal value of $k_{app}$ in the absence of ligand.

The CTPase activity of ParB was measured essentially as described for ParA, using reaction mixtures that contained 5 μM ParB, 1 mM CTP, 20 U/mL pyruvate kinase, 20 U/mL L-lactate dehydrogenase, 800 μg/mL NADH and 3 mM phosphorenolpyruvate in 200 μL ParA reaction buffer.

## Nucleotide content analysis

ParA or ParA-D60A (50 μM) were incubated alone, with a 1:20 molar excess of ParB$_{1-20}$ peptide (1 mM) and/or with 1 mg/mL non-specific (ns) DNA in reaction buffer (25 mM HEPES-KOH pH 7.4, 100 mM KCl, 10 mM MgSO$_4$, 1 mM DTT) for 20 min at room temperature in a final volume of 50 μL. The reactions were then incubated for an additional 10 min with 50 μL of high-capacity nickel-NTA beads (QIAGEN) pre-equilibrated in reaction buffer. The mixture was applied onto 700 μL spin columns (MoBiTec) containing 10 μM pore-size filters and centrifuged at $800 \times g$ for 2 min. The resin was washed three times with 500 μL of reaction buffer, and bound ParA was eluted with 50 μL of reaction buffer supplemented with 500 mM imidazole. To analyze the nucleotide content, 10-μL samples of the eluates from the affinity beads were mixed with 40 μL of double-distilled water. After the addition of 150 μL chloroform, the samples were vigorously agitated for 5 sec, heated for 15 s at 95 °C to denature proteins and release the bound nucleotides, and then snap-frozen in liquid nitrogen. After thawing of the samples, phase separation was accelerated by

centrifugation (17,300 × $g$, 10 min, 4 °C), and the aqueous phases containing the nucleotides were withdrawn. Nucleotide content was determined by high-performance liquid chromatography (HPLC) on an Agilent 1260 Infinity system equipped with a Metrosep A Supp 5–150/4.0 column (Metrohm). To this end, 10-μL samples of the aqueous phases obtained were injected into the column. Subsequently, nucleotides were separated with 100 mM $(NH_4)_2CO_3$ pH 9.25 at a flow rate of 0.6 mL/min and detected by photometry at a wavelength of 260 nm. Solutions of ADP and ATP (100 μM in double-distilled water; purity of ≥95%), analyzed in the same manner as the ParA/ParA$_{D60A}$-containing samples, served as standards for the identification of the protein-bound nucleotides.

### Nuclear magnetic resonance (NMR) spectroscopy

All NMR experiments were conducted at 298 K using a Bruker Avance III 800 MHz spectrometer equipped with a triple-resonance cryoprobe. To ensure buffer matching, the titrations were performed after dialysis of both $^2$H/$^{13}$C/$^{15}$N-labeled ParA$_{21-274}$-D60A and unlabeled ParB$_{1-20}$ against sample buffer, consisting of 50 mM HEPES (pH 6.8), 150 mM KCl, 5 mM MgSO$_4$, 1 mM DTT and 1 mM ATP. The initial concentration of ParA was determined to be 300 μM using ultraviolet absorbance at 280 nm and a predicted molar extinction coefficient of 13,410 M$^{-1}$ cm$^{-1}$ [106]. The NMR samples included 10% (v/v) D$_2$O as a locking agent. To reduce dilution effects, ParB$_{1-20}$ was gradually added in small increments from a stock solution of 7 mM to a final protein-to-peptide molar ratio of 1:5.8. Transverse relaxation-optimized spectroscopy (TROSY) $^1$H-$^{15}$N HSQC experiments were collected at each step using apodization weighted acquisitions[72,107,108], and the overlaid spectra are shown in Fig. 3a and Supplementary Fig. 8. Partial backbone assignments of ParA were obtained using TROSY versions of standard $^1$H-$^{13}$C-$^{15}$N scalar correlation experiments with $^2$H-decoupling[72,108]. The poor transverse relaxation properties of the ParA dimer, which has a size of ~60 kDa, resulted in weak carbon resonance signals, and we confidently assigned only ~80 out of 250 amide peaks. The raw NMR data were processed with NMRpipe[109]. Backbone chemical shifts of $^{13}$C$_\alpha$, $^{13}$C$_\beta$, $^{15}$N, and $^1$H$^N$ nuclei were assigned using CARA[110], while titration experiments were analyzed with NMRFAM-Sparky[111]. The chemical shift perturbations (CSPs) were calculated according to CSP = $[(0.14\Delta\delta N)^2 + (\Delta\delta H)^2]^{1/2}$, where $\Delta\delta N$ and $\Delta\delta H$ are the chemical shift changes of the $^{15}$N and $^1$H$^N$ nuclei, respectively, comparing ParA in the absence and presence of a 0.7-fold molar equivalent of ParB$_{1-20}$ (see also Fig. 3b). This point along the titration was used in our analysis to avoid the tracking of amide peaks that disappear over the course of the titration, as is characteristic in complexes exhibiting intermediate exchange (see also Supplementary Fig. 8).

### Hydrogen-deuterium exchange (HDX) mass spectrometry

HDX-MS analyses were conducted essentially as described previously[12] with minor modifications. Three different experiments were conducted (see also Supplementary Data 1): For Dataset 1, all samples contained 50 μM wild-type ParA and 1 mM ATP either without further additives (state 1; apo) or with 1 mg/mL salmon sperm DNA (state 2; DNA), 1 mM ParB$_{1-20}$ peptide (state 3; peptide) or both 1 mg/mL salmon sperm DNA and 1 mM ParB$_{1-20}$ peptide (state 4; DNA/peptide). For Dataset 2, all samples contained 50 μM ParA-D60A and 1 mM ATP either without further additives (state 1; apo) or with 1 mg/ml salmon sperm DNA (state 2; DNA), 1 mM ParB$_{1-20}$ peptide (state 3; peptide), or both 1 mg/mL salmon sperm DNA and 1 mM ParB$_{1-20}$ peptide (state 4, DNA/peptide). For Dataset 3, all samples contained 50 μM ParA-D60A, 1 mM ATP and 1 mM CTP either without further additives (state 1; apo) or with 1 mg/mL salmon sperm DNA (state 2; DNA), 100 μM ParB-Q52A and 2.5 μM parS-containing DNA stem-loop (parS-Mxan-wt) (state 3; ParB rings) or 1 mg/mL salmon sperm DNA, 100 μM ParB-Q52A and 2.5 μM parS-containing DNA stem-loop (state 4; DNA/ParB rings). Whenever included in the reaction, ParB-Q52A was pre-incubated with parS and CTP for 30 min at

ambient temperature to induce its transition to the closed (ring) state before it was mixed with ParA-D60A and ATP.

The preparation of the HDX reactions was aided by a two-arm robotic autosampler (LEAP Technologies). In short, 7.5 μl of sample was mixed with 67.5 μl of D$_2$O-containing buffer (25 mM HEPES-KOH pH 7.4, 10 mM MgSO$_4$, 100 mM KCl, 150 mM potassium glutamate, 1 mM DTT, 1 mM ATP; with 1 mM CTP supplemented for HDX dataset 3) to initiate the HDX reaction. After incubation for 10, 30, 100, 1000 or 10,000 s at 25 °C, 55-μl samples were taken from the reaction and mixed with an equal volume of quench buffer (1.2 M KH$_2$PO$_4$/H$_3$PO$_4$, 2 M guanidine-HCl, pH 2.2) precooled to 1 °C. Ninety-five microliters of the resulting mixtures were immediately injected into an ACQUITY UPLC M-Class System with HDX Technology (Waters)[112]. Undeuterated HDX reactions were prepared similarly by 10-fold dilution in H$_2$O-containing buffer. The injected samples were flushed out of the loop (50 μl) with H$_2$O + 0.1% (v/v) formic acid (100 μl/min flow rate) over 3 min and guided to a column (2 mm × 2 cm) that was packed with immobilized porcine pepsin and cooled to 12 °C for proteolytic digestion. The resulting peptic peptides were collected over the three-minute period of sample injection and digestion on an ACQUITY UPLC BEH C18 VanGuard Pre-column (130 Å, 1.7 μm, 2.1 mm × 5 mm; Waters) that was kept at 0.5 °C. Afterwards, the trap column was placed in line with an ACQUITY UPLC BEH C18 1.7 μm 1.0 × 100 mm column (Waters), and the peptic peptides were separated at 0.5 °C using a gradient of water + 0.1% (v/v) formic acid (A) and acetonitrile + 0.1% (v/v) formic acid (B) at a flow rate of 60 μl/min as follows: 0-7 min/95–65% A, 7–8 min/65-15% A, 8–10 min/15% A, 10–11 min/5% A, 11–16 min/95% A. Peptides were ionized with an electrospray ionization source (250 °C capillary temperature, 3.0 kV spray voltage) and mass spectra were acquired in positive ion mode over a range of 50–2000 $m/z$ on a G2-Si HDMS mass spectrometer with ion mobility separation (Waters), using Enhanced High Definition MS (HDMS$^E$) or High Definition MS (HDMS) mode for undeuterated and deuterated samples, respectively[113,114]. Lock mass correction was implemented with a [Glu1]-Fibrinopeptide B standard (Waters). During peptide separation, the loop and pepsin column were washed three times with 80 μL each of 4% (v/v) acetonitrile and 0.5 M guanidine-HCl. Blank injections were performed between each sample. All measurements were performed in triplicate as independent HDX reactions.

ParA and ParA$_{D60A}$ peptides were identified from the undeuterated samples acquired with HDMS$^E$ with the software ProteinLynx Global SERVER version 3.0.1 (PLGS; Waters), using low energy, elevated energy and intensity thresholds of 300, 100, and 1000 counts, respectively. Ions were matched to peptides using a database containing the amino acid sequences of ParA or ParA$_{D60A}$ and their respective reversed sequences with search parameters as described previously[9]. After automated data processing with the software DynamX version 3.0.0 (Waters), all spectra were inspected manually, and peptides were omitted if necessary (e.g. in case of a low signal-to-noise ratio or the presence of overlapping peptides). An overview of dataset parameters and characteristics of the HDX-MS experiments is given in Supplementary Data 1.

### Widefield fluorescence microscopy

The *M. xanthus* strains analyzed carried the endogenous *parB* gene under the control of the copper-inducible P$_{cuo}$ promoter and the indicated *sfmTurqoise2$^{ox}$-parB* alleles under the control of the vanillate-inducible P$_{van}$ promoter. The cells were grown in CTT medium containing 300 μM CuSO$_4$, washed three times with fresh CTT medium, and then cultivated for 20 h in CTT medium supplemented with 500 μM sodium vanillate to deplete wild-type ParB and produce the sfmTurqoise2$^{ox}$-tagged variants instead. Subsequently, they were spotted on 1.5% (w/v) agarose pads prepared in TPM buffer (10 mM Tris-HCl, 8 mM MgSO$_4$, 1 mM potassium phosphate, pH 7.6) supplemented with 10% (v/v) CTT medium[115]. Images were taken with a Zeiss

Axio Imager.Z1 microscope equipped with a 3 100/1.46 Oil DIC objective and a pco.edge 4.2 sCMOS camera (PCO). An X-Cite 120PC metal halide light source (EXFO, Canada) and an ET-CFP filter cube (Chroma, USA) were used for the detection of fluorescence signals. Images were recorded with VisiView 4.0.0.14 (Visitron Systems, Germany) and processed with Fiji 1.49[116]. The subcellular localization of fluorescence signals was analyzed with BacStalk[117]. FRAP analysis was performed with the Zeiss Axio. Observer Z1 setup described above, using a 488 nm-solid state laser and a 2D-VisiFRAP multi-point FRAP module (Visitron Systems, Germany), with 2-ms pulses at a laser power of 30%. After the acquisition of a prebleach image and the application of a laser pulse, 14 images were taken at 10 s intervals. For each time point, the integrated fluorescence intensities of an unbleached whole cell, the bleached region and an equally sized unbleached region were measured using Fiji 1.49[116]. After correction for photobleaching, recovery half-times were calculated independently for each cell by fitting the data to a single-exponential function using the Solver add-in of Microsoft Excel 2019.

## Immunoblot analysis

The *M. xanthus* strains analyzed were grown as described for widefield fluorescence microscopy. Cells from 1.5-mL samples of the cultures were harvested by centrifugation, resuspended in SDS sample buffer to a theoretical $OD_{550}$ of 10 and then incubated for 15 min at 98 °C. Proteins were separated in a 15% SDS-polyacrylamide gel and transferred to a polyvinylidene fluoride (PVDF) membrane (Immobilon; Millipore). Immunoblot analysis was conducted as described previously[23], using a polyclonal anti-GFP antibody (Sigma, Germany; Cat. #: G1544; RRID: AB_439690) at a dilution of 1:10,000. Goat anti-rabbit immunoglobulin G conjugated with horseradish peroxidase (Perkin Elmer, USA) was used as a secondary antibody. Immunocomplexes were detected with the Western Lightning Plus-ECL chemiluminescence reagent (Perkin Elmer, USA). The signals were recorded with a ChemiDoc MP imaging system (BioRad, Germany) and analyzed using Image Lab software (BioRad, Germany).

## Bioinformatic analysis

The molecular structures of the complexes formed between *M. xanthus* ParA and the full-length ParB protein or the ParB$_{1-20}$ peptide were predicted using ColabFold (Alphafold-Multimer)[70,118] with 24 recycles and amber relaxation enabled. Protein structures were processed with UCSF ChimeraX v1.5[103]. Protein similarity searches were performed with BLAST[119], using the BLAST server of the National Institutes of Health (https://blast.ncbi.nlm.nih.gov/Blast.cgi). Sequence alignments were generated with Clustal Omega[120] and processed with Jalview[121] and WebLogo[122].

## Statistics and reproducibility

All experiments were performed at least twice with similar results. No data were excluded from the analyses. Details on the number of replicates and the sample sizes are given in the figure legends. Standard deviations were calculated in Microsoft Excel 2019. Curve fitting was performed using the Solver function of Microsoft Excel 365. To quantify imaging data, multiple randomly selected fields of cells were analyzed per condition. The analyses included all cells in the micrographs that were clearly separable from each other.

## Availability of biological material

All plasmids and strains generated in this study are available from the corresponding author without restriction.

## Reporting summary

Further information on research design is available in the Nature Portfolio Reporting Summary linked to this article.

## Data availability

The coordinates and structure factors for the crystal structure of the His$_6$-ParA$_{21-274}$-D60A•ATP dimer were deposited at the RCSB Protein Data Bank (PDB)[123] under the accession code 8RAY. The PDB accession codes for the published crystal structures referenced in this study are: 6IUD, 2BEK, 7NPD, 2OZE, 4E09, 7DV3, 5K5Z, 3Q9L, and 3R9J. The assigned NMR chemical shifts of His$_6$-ParA$_{21-274}$-D60A have been deposited to the Biological Magnetic Resonance Data Bank (bmrb.io)[124] under accession number 52940. HDX-MS data have been deposited to the ProteomeXchange Consortium via the PRIDE[125] partner repository under the dataset identifier PXD063735. All other data supporting the findings of this study are included in the main text or the supplementary material. Source data are provided with this paper.

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

## Acknowledgements

We thank Julia Rosum for excellent technical assistance. This work was supported by the University of Marburg (core funding to M.T.), the Max Planck Society (Max Planck Fellowships to M.T. and G.B.), the German Research Foundation (project 505997786— GRK 2937 to M.T. and G.B., project 260989694 to G.B. and project 324652314 to G.B.), and the European Research Council (grant agreement 101097986—C-SWITCH to M.T.). C.P.-B. was supported by an EMBL Interdisciplinary Postdoc (EI₃POD) Program fellowship under Marie Sklodowska-Curie Actions COFUND (grant no. 664726). J.H. gratefully acknowledges support by the European Molecular Biology Laboratory (EMBL).

## Author contributions

L.S. and M.O.-V. contributed equally and should be regarded as joint first authors. C.P.-B. and W.S. also contributed equally and should be considered joint second authors. L.S., M.O.-V., and M.T. conceived the study. L.S. and M.O.-V. constructed plasmids and strains and conducted the biochemical and in silico studies. J.Ha. and M.Tham constructed plasmids and performed initial biochemical experiments. L.S. conducted the cell biological studies. L.S., M.O.-V., and M.T. analyzed the biochemical and cell biological data. B.S. and C.P.-B. performed the NMR experiments. C.P.-B. and J.He. analyzed the NMR data. W.S. performed the HDX and HPLC analyses. C.-N.M. performed the crystallization screens and solved the molecular structure of ParA. J.H., G.B., and M.T. secured funding and supervised the study. L.S. and M.T. wrote the manuscript, with input from all other authors.

## Funding

## Competing interests

The authors declare no competing interests.
