## [Transparent Peer Review file · Nature Communications]

Molecular basis of ParA ATPase activation by the CTPase ParB during bacterial chromosome segregation

Corresponding Author: Professor Martin Thanbichler

Version 0:

Reviewer comments:

Reviewer #1

(Remarks to the Author)

The manuscript by Schnabel and co-workers presents an interesting set of results that provide new mechanistic insights into the bacterial ParABS DNA partitioning system. The main focus is ParA from the model organism *Myxococcus xanthus* and its interactions with ParB and DNA that activate the ParA ATPase. Using a combination of x-ray crystallography, protein NMR, HDX mapping, modelling, cell biology, binding and kinetic assays, a compelling model of the ParA-ParB complex is developed, with identification of residues critical for the activation of ParA. A cohesive and extensive collection of data supports this model for the interaction and suggests that ParB binding alters the network of interactions around the ParA catalytic site to influence hydrolytic activity. Although a detailed atomic-level view of the mechanism is not possible to elucidate from the data, ParA binding cooperativity for DNA and ParB is demonstrated, along with subtle changes in conformation and/or dynamics around the catalytic site induced by ParB and nsDNA binding together.

The following are minor issues recommended to be addressed in this work:

1) Binding versus ATPase activity

Page 9, line 211 - Binding of ParA to serine point mutations in ParB clamps in BLI experiments was said to be 'abolished', and the effect of some ParA mutations in were said to 'completely block' the interaction. These are strong statements given that only one concentration of ParA was tested. The concentration at which this difference in binding was observed should be stated since a quantitative measure of the affinity was not performed. Also, the impact of ParA mutations on its structure were not evaluated – evidence that the structures were preserved should be provided.

In general it is difficult to compare the effect of mutations on binding versus stimulation of ATPase activity (i.e. Fig 5 a, b and Fig 6 d, e). Only one (unspecified) concentration of protein was evaluated in binding assays which prevents a quantitative comparison of the impact of the mutation on binding affinity. In Fig. 6 e) only one ParB concentration was tested in ATPase assays – this makes it impossible to know if the apparent K_d for ParB, and/or the maximal k_{cat} values were affected by the mutation. If apparent K_d values for mutants are not available, additional information needs to be provided in the manuscript that can either justify these comparisons or provide context for these comparisons that acknowledges these possibilities.

2) Kinetic assay results

More information is needed about the kinetic measurements of ATPase activity. Examples of raw data and its conversion into k_{cat} values is missing. Since k_{cat} can be difficult to measure if ATP affinity is reduced by a mutation, some description of k_{cat} measurements should be provided in the Supplementary Material.

The equations used to fit kinetic profiles in Figure 1f) need to be provided. If cooperativity was included in the fit the hill coefficient should also be reported. Also, what is the basal activity obtained with $\Delta 21$ coming from? If non-zero basal activity exists then it does not make sense for fits to go to k_{cat} values of zero at $[ParB] = 0$.

ATP hydrolysis rates are slow but may be consistent with other members of the ParB family. It would be helpful to provide some frame of reference by comparison of measured catalytic constants to what would be expected for members of this family of ATPases.

3) Sequence numbering

The numbering of residues in the crystal structure does not match the numbering used in the paper. The numbering of residues in the deposited chemical shifts also differs from both the structure and the manuscript. This needs to be harmonized.

4) Atomic coordinates for the model

It would be helpful for readers to have access to the atomic coordinates for this model. Without this model it was not possible to determine if the region of ParA indicated as the 'interface' between DNA binding and ParB binding (residues 167 – 179) make any interactions with ParB in the model. Also, the colour scale for pLDDT scores in Supplementary Figure 6 needs to be included.

5) Changes in conformation versus dynamics

In the abstract and discussion of results, it is suggested that the data shows that there are conformational changes induced in the Walker motifs by ParB binding. Although the exchange data are consistent with this proposal, it should also be acknowledged that the observed changes in HDX patterns may also arise due to a change in conformational dynamics.

6) Comparison with MinD/E

On page 18, line 480 – 481 there is a statement made regarding the interaction between the E. coli MinD and MinE proteins that needs some correction. An N-terminal helix from MinE binds the MinD dimer interface, and it seems like there might be some overlap with the ParB-ParA interaction site, particularly if the structure of the complex with full-length MinE is considered. (Again, it would be helpful to have coordinates to their model to see this.)

7) Comment on HDX results (not a critique...)

It is surprising that the HDX data does not show a change in exchange levels for the D60A mutant in the presence and absence of ParB for more residues around the putative interaction site. Although this is attributed to the low affinity of the ParB-ParA interaction, there was a subtle increase in exchange in residues C-terminal to the Walker B motif, one of which also showed large chemical shift changes upon ParB binding. This observation paints an intriguing picture of ParB interactions with residues Leu150, Gly151, and Leu152 that are transmitted into conformational and/or dynamics changes in the adjacent Walker motifs to promote ATP hydrolysis.

It is not clear why other residues that are also predicted to line the ParB-binding groove do not show any change in HDX exchange, but it is possible that more differences would have been detected if exchange had been allowed to proceed for more than the 2.8 hour maximum used in this study, particularly since exchange in these regions was not detected with shorter incubation times. However, it is also possible that increased protection from solvent exchange by ParB binding could be masked by increased dynamics caused by this binding event. Consequently this is not an experiment that is being requested, unless said data already happens to be available...!

Reviewer #2

(Remarks to the Author)

In this paper the authors elegantly dissect how interactions between the ParA and ParB proteins in the ParABS system function in DNA recognition and processing. Unknowns in the field were knowledge of the interaction interface between the two protein and how they modulated DNA binding. Towards answering this question the team has used BLI, HDX-MS, NMR, crystallography, and in vivo functional studies to arrive at an interaction interface.

All the experiments are well done and interpreted appropriately. The conclusions align with the results and the manuscript is well-written and easy to read. This reviewer particularly appreciated the historic context and how the questions were setup.

From the HDX-MS analysis perspective, the data look good. The interpretations are appropriate, and there are no major concerns. One aspect could be refined a bit more: The authors do these experiments with the peptide and to induce the changes. Can these be done with the full-length proteins? In the results, the authors switch to using ParB as a general term for explaining the results. It would be more appropriate to say ParB-peptide to be more accurate.

Both concentration dependent data in Figure 2 and supplemental Figures show cooperative properties; Yet, the data are fit to a single-site model. Would the data not be better described by a two-site model? These experiments all point to some form of cooperativity-based binding mechanism and these need to be discussed as a possibility.

Reviewer #3

(Remarks to the Author)

The ParABS system is conserved in bacteria and positions chromosomal origins, plasmids, and even non-DNA cargos such as carboxysomes. Its two core proteins, ParA and ParB, both dimerize, bind and hydrolyse nucleotides (ATP and CTP, respectively), bind DNA (ParA non-specifically, ParB to parS), and interact with each other. Years of work have linked these properties to sophisticated models of chromosome origin segregation, in which iterations of nucleotide binding/hydrolysis and mutual interaction determine ParA and ParB monomeric/dimeric state, driving their relocation within the cell and the dynamic positioning of the centromere (ParB nucleoprotein complex).

Building on the recent discovery that CTP loading upon parS-binding closes the ParB dimer into a clamp able to slide along flanking DNA - explaining the formation of a nucleoprotein complex, the present study provides the missing mechanistic link between ParB clamp formation and ParA activity. Using *Myxococcus xanthus* - a species with well-characterized

chromosome segregation dynamics and ParB biochemistry - the authors combine in vitro approaches assessing DNA and protein interactions (biolayer interferometry) and ATPase activity, with structural modeling, and validation through targeted mutagenesis, NMR, HDX, and in vivo assays (subcellular localization, FRAP), to comprehensively dissect the ParA-ParB interface.

Major findings include:

- 1) ParA preferentially binds the closed ParB clamp, confining their interaction at the centromere despite ParA non-specific DNA binding across the nucleoid;
- 2) DNA and ParB binding cooperatively stimulate ParA ATPase activity;
- 3) Key contact residues between both proteins are confirmed or identified. Data allow the mapping of the subtle ParA conformational changes and impact on ATPase activity that these residues trigger, providing mechanistic, structural insights into ParA function.

In light of the recent breakthrough of the ParB CTPase cycle, these findings substantially refine our understanding of how the ParABS system drives active chromosome segregation. This study is of broad interest because the ParABS module is highly conserved. The discussion nicely highlights the conservation or potential adaptations of residues uncovered here, offering clear directions for future studies in other species or related systems beyond chromosome partitioning.

The work is carefully conducted, the manuscript is clearly written, and figures are excellent. I only have minor comments:

- Line 25: "a ParB dimer"
- Lines 60-62: this statement is not universal; in some species, multiple ParABS-dependent segregation rounds occur concomitantly along several ParA gradients (Kaljevic et al, Curr Biol 2021; Plos Genetics 2023)
- Line 81: "governs"
- Line 106: "hydrolyzable"
- Lines 155-159: Before concluding that ParB clamp closure is required for ParA-ParB interaction, it would help to include a brief, more direct conclusion about the assay, which measures ParA ATPase activity rather than direct ParA-ParB binding.
- Lines 180-182: Clarify the link between ParA "nucleoid-bound state" and the findings presented in this section (ParB and DNA binding cooperatively activate ParA's ATPase activity).
- Line 395 : "abovementioned"
- Line 1109 : "represent"
- Line 1122 : "was loaded"
- Line 1161 : « representation »
- SFig2 legend: "representative experiment"
- SFig3a legend: remove the duplicated anti-GFP antibody information.

Version 1:

Reviewer comments:

Reviewer #1

(Remarks to the Author)

Appropriate corrections and responses to reviewers' comments were included in the updated manuscript. These include new data added to the Supplementary Material confirming the structural integrity of the ParA* mutants. Additional kinetics data were also acquired that show that mutants have an impact on binding and activation of catalysis. A supplementary figure has been added to visualize a comparison of ParA/B versus MinD/E complexes, which helps to highlight the differences. The authors also took care to harmonize the sequence numbering between paper, structure and chemical shift files.

The corrections strengthen what was already a strong complement of data in this interesting paper that shows how ParA interactions with DNA and ParB give rise to conformational changes that activate its ATPase.

Reviewer #2

(Remarks to the Author)

The authors have adequately addressed my concerns.

Reviewer #3

(Remarks to the Author)

The authors convincingly addressed all my comments.

Reviewer #1

The manuscript by Schnabel and co-workers presents an interesting set of results that provide new mechanistic insights into the bacterial ParABS DNA partitioning system. The main focus is ParA from the model organism *Myxococcus xanthus* and its interactions with ParB and DNA that activate the ParA ATPase. Using a combination of x-ray crystallography, protein NMR, HDX mapping, modelling, cell biology, binding and kinetic assays, a compelling model of the ParA-ParB complex is developed, with identification of residues critical for the activation of ParA. A cohesive and extensive collection of data supports this model for the interaction and suggests that ParB binding alters the network of interactions around the ParA catalytic site to influence hydrolytic activity. Although a detailed atomic-level view of the mechanism is not possible to elucidate from the data, ParA binding cooperativity for DNA and ParB is demonstrated, along with subtle changes in conformation and/or dynamics around the catalytic site induced by ParB and nsDNA binding together.

The following are minor issues recommended to be addressed in this work:

1) Binding versus ATPase activity

Page 9, line 211 - Binding of ParA to serine point mutations in ParB clamps in BLI experiments was said to be 'abolished', and the effect of some ParA mutations in were said to 'completely block' the interaction. These are strong statements given that only one concentration of ParA was tested. The concentration at which this difference in binding was observed should be stated since a quantitative measure of the affinity was not performed. Also, the impact of ParA mutations on its structure were not evaluated – evidence that the structures were preserved should be provided.

We are sorry for not providing the protein concentration used in the binding assays. The reactions contained ParA* or its mutant variants at a concentration of 5 μ M, which was sufficient to observe considerable binding of wild-type ParA* to model partition complexes (Figure 1b). A comparison of the binding curves in Figure 1b with those in Figures 2h and 2j indicates that the mutant proteins have hardly any or no binding activity under the same conditions. To account for the fact that the binding affinities were not determined quantitatively by assaying a wider range of protein concentrations, we now state that the interactions were "strongly impaired" and "largely abolished".

To assess the effect of the amino acid substitutions on the structure of ParA, we re-purified three representative ParA* variants with strongly reduced ParB binding affinity and analyzed their ability to form ATP-bound dimers by gel filtration chromatography (new Supplementary figure 7). The behavior of the three proteins was virtually undistinguishable from that of the wild-type protein, verifying that they did not have any significant defects in ATP binding and dimerization.

In general it is difficult to compare the effect of mutations on binding versus stimulation of ATPase activity (i.e. Fig 5 a, b and Fig 6 d, e). Only one (unspecified) concentration of protein was evaluated in binding assays which prevents a quantitative comparison of the impact of the mutation on binding affinity. In Fig. 6 e) only one ParB concentration was tested in ATPase assays – this makes it impossible to know if the apparent K_d for ParB, and/or the maximal k_{cat} values were affected by the mutation. If apparent K_d values for mutants are not available, additional

information needs to be provided in the manuscript that can either justify these comparisons or provide context for these comparisons that acknowledges these possibilities.

We are sorry for not providing the ParA* concentrations used in the BLI assays (5 μ M). This information is now provided in the legends to Figures 5 and 6.

We agree that the defects caused by mutations in the helix H4/h6/H7 region are likely due to a decrease in both the binding affinity (K_D) and the catalytic activity (v_{max}). Therefore, we now conclude that the *in vitro* and *in vivo* results show “that the positively charged residue at the center of the ParA-binding motif critically contributes to ParB function, both by supporting the recruitment of ParA and mediating the stimulatory effect of ParB on its ATPase activity”.

To further investigate this point, we re-purified the Y81L and E117R variants and determined their ATPase activities over a range of ParB concentrations (new Supplementary figure 11a). The results obtained suggest that both their K_D and v_{max} values are lower than those measured for the wild-type protein, indicating a reduced binding affinity for ParB and a reduced sensitivity to the stimulatory effect of ParB.

We have now also performed ATPase assays for the Y81L and E117R variants at elevated ATP concentrations (5 mM and 10 mM). In these conditions, the mutant proteins still show similar defects in their sensitivity to ParB, suggesting that their reduced ATPase activities are not caused by a reduction in their affinity for ATP (new Supplementary figure 11b).

2) Kinetic assay results

More information is needed about the kinetic measurements of ATPase activity. Examples of raw data and its conversion into k_{cat} values is missing. Since k_{cat} can be difficult to measure if ATP affinity is reduced by a mutation, some description of k_{cat} measurements should be provided in the Supplementary Material.

The raw data and their analysis are now provided in the Source Data file. All ATPase assays were performed with an ATP concentration of 1 mM, which is sufficient to largely saturate ParA with nucleotide. However, we agree that the determination of precise k_{cat} values would require titration experiments. We have therefore changed “ k_{cat} ” to “ k_{app} ”, which represents the apparent turnover number under the given conditions.

The equations used to fit kinetic profiles in Figure 1f) need to be provided. If cooperativity was included in the fit the hill coefficient should also be reported. Also, what is the basal activity obtained with delta21 coming from? If non-zero basal activity exists then it does not make sense for fits to go to k_{cat} values of zero at $[ParB] = 0$.

We have now fitted all data to a Hill equation and described the approach used in the Methods section. However, for the first data points, the concentrations of ParB are within the range of the ParA concentration used in the reaction and close to the K_D value of the ParA-ParB interaction (see Figure 1b). Moreover, due to the close linkage of the two N-terminal peptides in the ParB dimer, the binding reactions may be influenced by avidity effects. Therefore, it is not straightforward to determine whether the sigmoidal shape of the binding curve really indicates cooperativity in the binding process. We would thus prefer not to give a Hill coefficient in order

to prevent readers from drawing potentially incorrect conclusions. We have now added a note to the legend for Figure 1f describing this caveat.

The data shown in the previous Figure 1f were from an early phase of the project in which our ParA preparations still contained traces of contaminating ATPases, leading to a low level of background ATPase activity in our assays. We have now repeated the measurements with newly purified protein and no longer observed any significant background activity.

ATP hydrolysis rates are slow but may be consistent with other members of the ParB family. It would be helpful to provide some frame of reference by comparison of measured catalytic constants to what would be expected for members of this family of ATPases.

Thank you for raising this point. Since the ATPase activity of ParA remains low, even when stimulated by both ParB and DNA, providing references to values determined for other systems may indeed be helpful. Typically, the maximal turnover numbers reported for other ParA proteins are around 1 min^{-1} . We now state in the Results section that the “maximal turnover rates obtained were similar to those reported for other ParA orthologs (Leonard et al, 2005; Lim et al, 2014; Taylor et al, 2021).”

3) Sequence numbering

The numbering of residues in the crystal structure does not match the numbering used in the paper. The numbering of residues in the deposited chemical shifts also differs from both the structure and the manuscript. This needs to be harmonized.

We have now updated the deposited crystal structure and chemical shift data to harmonize the numbering of the residues.

4) Atomic coordinates for the model

It would be helpful for readers to have access to the atomic coordinates for this model. Without this model it was not possible to determine if the region of ParA indicated as the ‘interface’ between DNA binding and ParB binding (residues 167 – 179) make any interactions with ParB in the model. Also, the colour scale for pLDDT scores in Supplementary Figure 6 needs to be included.

We have now included the atomic coordinates of the model (new Supplementary file 1).

The label ‘Interface’ does not refer to a region connecting the ParB- and DNA-binding sites but to a region at the dimer interface that connects the DNA-binding site with the Walker A loop. The figure legend has been amended to make this clearer. Moreover, we have added a color scale for the pLDDT scores in Supplementary figure 6.

5) Changes in conformation versus dynamics

In the abstract and discussion of results, it is suggested that the data shows that there are conformational changes induced in the Walker motifs by ParB binding. Although the exchange data are consistent with this proposal, it should also be acknowledged that the observed changes in HDX patterns may also arise due to a change in conformational dynamics.

We now state in the Discussion that “act synergistically to alter the structure and/or conformational dynamics of the Walker A and Walker B regions in the ParA dimer”.

6) Comparison with MinD/E

On page 18, line 480 – 481 there is a statement made regarding the interaction between the E. coli MinD and MinE proteins that needs some correction. An N-terminal helix from MinE binds the MinD dimer interface, and it seems like there might be some overlap with the ParB-ParA interaction site, particularly if the structure of the complex with full-length MinE is considered. (Again, it would be helpful to have coordinates to their model to see this.)

We have now included an additional supplementary figure comparing the binding sites of the ParB₁₋₂₀ peptide and full-length MinE on their corresponding ATPases (new Supplementary Figure 15). It is true that both effectors associate with the same face of the interacting ATPase dimer. However, their actual binding sites are distal from each other, with ParB₁₋₂₀ and MinE only overlapping in regions that are not involved in the binding reaction. The new figure also shows more clearly now that the helix H4/H6/H7 region of ParA, which is critically involved in the interaction with ParB₁₋₂₀, is not conserved in MinD.

7) Comment on HDX results (not a critique...)

It is surprising that the HDX data does not show a change in exchange levels for the D60A mutant in the presence and absence of ParB for more residues around the putative interaction site. Although this is attributed to the low affinity of the ParB-ParA interaction, there was a subtle increase in exchange in residues C-terminal to the Walker B motif, one of which also showed large chemical shift changes upon ParB binding. This observation paints an intriguing picture of ParB interactions with residues Leu150, Gly151, and Leu152 that are transmitted into conformational and/or dynamics changes in the adjacent Walker motifs to promote ATP hydrolysis. It is not clear why other residues that are also predicted to line the ParB-binding groove do not show any change in HDX exchange, but it is possible that more differences would have been detected if exchange had been allowed to proceed for more than the 2.8 hour maximum used in this study, particularly since exchange in these regions was not detected with shorter incubation times. However, it is also possible that increased protection from solvent exchange by ParB binding could be masked by increased dynamics caused by this binding event. Consequently this is not an experiment that is being requested, unless said data already happens to be available...!

We routinely sample HDX reactions from 10 to 10,000 seconds (2.8 h). We agree that this time frame may not be ideal from the HDX perspective, since a larger sampling window always provides more sensitivity. However, it was advantageous in terms of protein stability to avoid excessively long incubation times and also practically more feasible to operate within this time range. Therefore, unfortunately, HDX data collected after more than 10,000 sec of incubation are not available.

An analysis of the HDX data obtained based on the model of the ParA-ParB complex indicates that the lack of obvious differences at the postulated interaction site is not explained by an exceedingly low HDX rate in this protein region. The overall HDX observed in our dataset ranges between 60 and 65%, suggesting that back-exchange was under reasonable control during the experiments. The HDX measured in regions of ParA lining the postulated ParB binding site were ~20-25% (at

10,000 s) for residues 110-117, ~30% for residues 149-152 and ~40-50% for residues 175-193 (except for the residue 183-187 patch). To us, these values appear to be within a range in which a reduction in HDX upon ParB binding (if present) should have been observable. Thus, HDX-MS did not appear to be suitable for resolving ParB binding, although it has proven to be a valuable approach to resolving transient interactions in other systems. For this reason, we turned to NMR to gain further insight into the ParB binding site.

Reviewer #2

In this paper the authors elegantly dissect how interactions between the ParA and ParB proteins in the ParABS system function in DNA recognition and processing. Unknowns in the field were knowledge of the interaction interface between the two protein and how they modulated DNA binding. Towards answering this question the team has used BLI, HDX-MS, NMR, crystallography, and in vivo functional studies to arrive at an interaction interface.

All the experiments are well done and interpreted appropriately. The conclusions align with the results and the manuscript is well-written and easy to read. This reviewer particularly appreciated the historic context and how the questions were setup.

From the HDX-MS analysis perspective, the data look good. The interpretations are appropriate, and there are no major concerns. One aspect could be refined a bit more: The authors do these experiments with the peptide and to induce the changes. Can these be done with the full-length proteins? In the results, the authors switch to using ParB as a general term for explaining the results. It would be more appropriate to say ParB-peptide to be more accurate.

We performed three sets of HDX experiments with different sample compositions (see Supplementary Data 1, Supplementary Figure 9 and Figure 4).

Dataset 1: ParA-WT ± salmon sperm DNA ± ParB₁₋₂₀ peptide

Dataset 2: ParA-D60A ± salmon sperm DNA ± ParB₁₋₂₀ peptide

Dataset 3: ParA-D60A ± salmon sperm DNA ± ParB rings (full-length ParB-Q52A closed by preincubation with CTP and *parS* DNA)

The results obtained for the interaction of ParA-D60A with the ParB₁₋₂₀ peptide (Dataset 2) or full-length ParB rings (Dataset 3) are very similar, indicating that the ParB₁₋₂₀ peptide was a suitable proxy for full-length ParB in our study.

Both concentration dependent data in Figure 2 and supplemental Figures show cooperative properties; Yet, the data are fit to a single-site model. Would the data not be better described by a two-site model? These experiments all point to some form of cooperativity-based binding mechanism and these need to be discussed as a possibility.

We have now fitted all data to a Hill equation to account for the sigmoidal shape of the binding curves. However, as explained in our response to Reviewer #1 (point 2), there are some issues that make it uncertain whether the sigmoidal binding curves observed really reflect a cooperativity-based binding mechanism. We now state these issues in the legend to Figure 1f. However, we do mention the possibility of cooperative interactions between the two ParB-binding sites of

a ParA dimer in the Discussion and suggest residue Q62 as a potential mediator of these interactions.

Reviewer #3

The ParABS system is conserved in bacteria and positions chromosomal origins, plasmids, and even non-DNA cargos such as carboxysomes. Its two core proteins, ParA and ParB, both dimerize, bind and hydrolyse nucleotides (ATP and CTP, respectively), bind DNA (ParA non-specifically, ParB to parS), and interact with each other. Years of work have linked these properties to sophisticated models of chromosome origin segregation, in which iterations of nucleotide binding/hydrolysis and mutual interaction determine ParA and ParB monomeric/dimeric state, driving their relocation within the cell and the dynamic positioning of the centromere (ParB nucleoprotein complex).

Building on the recent discovery that CTP loading upon parS-binding closes the ParB dimer into a clamp able to slide along flanking DNA - explaining the formation of a nucleoprotein complex, the present study provides the missing mechanistic link between ParB clamp formation and ParA activity. Using *Myxococcus xanthus* - a species with well-characterized chromosome segregation dynamics and ParB biochemistry - the authors combine in vitro approaches assessing DNA and protein interactions (biolayer interferometry) and ATPase activity, with structural modeling, and validation through targeted mutagenesis, NMR, HDX, and in vivo assays (subcellular localization, FRAP), to comprehensively dissect the ParA-ParB interface.

Major findings include:

- 1) ParA preferentially binds the closed ParB clamp, confining their interaction at the centromere despite ParA non-specific DNA binding across the nucleoid;
- 2) DNA and ParB binding cooperatively stimulate ParA ATPase activity;
- 3) Key contact residues between both proteins are confirmed or identified. Data allow the mapping of the subtle ParA conformational changes and impact on ATPase activity that these residues trigger, providing mechanistic, structural insights into ParA function.

In light of the recent breakthrough of the ParB CTPase cycle, these findings substantially refine our understanding of how the ParABS system drives active chromosome segregation. This study is of broad interest because the ParABS module is highly conserved. The discussion nicely highlights the conservation or potential adaptations of residues uncovered here, offering clear directions for future studies in other species or related systems beyond chromosome partitioning.

The work is carefully conducted, the manuscript is clearly written, and figures are excellent. I only have minor comments:

- Line 25: "a ParB dimer"

Done.

- Lines 60-62: this statement is not universal; in some species, multiple ParABS-dependent segregation rounds occur concomitantly along several ParA gradients (Kaljevic et al, Curr Biol 2021; Plos Genetics 2023)

To avoid a further expansion of the Introduction, we now state that chromosome partitioning “typically” involves a single round of sister origin segregation along a nucleoid-spanning ParA gradient.

- Line 81: “governs”

Done.

- Line 106: “hydrolyzable”

Done.

- Lines 155-159: Before concluding that ParB clamp closure is required for ParA-ParB interaction, it would help to include a brief, more direct conclusion about the assay, which measures ParA ATPase activity rather than direct ParA-ParB binding.

We state that clamp closure is required for a *productive* interaction between ParB and ParA, which includes both the actual binding event and the stimulation of the ParA ATPase activity. Since the ATPase assay cannot differentiate between these two components, we would like to adhere to this terminology. We think that the direct conclusion that clamp closure is required for ParB to efficiently stimulate the ParA ATPase activity is already evident from the description of the data.

- Lines 180-182: Clarify the link between ParA “nucleoid-bound state” and the findings presented in this section (ParB and DNA binding cooperatively activate ParA’s ATPase activity).

We agree that this conclusion is not immediately obvious, although the observed cooperativity of ParB and DNA binding means that the two binding events stimulate each other reciprocally. We now conclude “that ParB stimulates both the DNA-binding and ATPase activities of ParA. In this way, it may reinforce the ParA-mediated tethers between partition complexes and the nucleoid, while also priming these tethering complexes for disassembly.”

- Line 395 : “abovementioned”

Done.

- Line 1109 : “represent”

Done.

- Line 1122 : “was loaded”

Done.

- Line 1161 : « representation »

Done.

- SFig2 legend: “representative experiment”

Done.

- SFig3a legend: remove the duplicated anti-GFP antibody information.

Done.

We would like to thank the three reviewers for the time and effort they invested in reviewing our manuscript and for their constructive criticism, which helped to significantly improve the paper.

We are happy that our revisions could address all the concerns raised.

Reviewer #1

Appropriate corrections and responses to reviewers' comments were included in the updated manuscript. These include new data added to the Supplementary Material confirming the structural integrity of the ParA* mutants. Additional kinetics data were also acquired that show that mutants have an impact on binding and activation of catalysis. A supplementary figure has been added to visualize a comparison of ParA/B versus MinD/E complexes, which helps to highlight the differences. The authors also took care to harmonize the sequence numbering between paper, structure and chemical shift files.

The corrections strengthen what was already a strong complement of data in this interesting paper that shows how ParA interactions with DNA and ParB give rise to conformational changes that activate its ATPase.

Reviewer #2

The authors have adequately addressed my concerns.

Reviewer #3

The authors convincingly addressed all my comments.